# Long Non-Coding RNAs in Drug Resistance of Gastric Cancer: Complex Mechanisms and Potential Clinical Applications

**DOI:** 10.3390/biom14060608

**Published:** 2024-05-22

**Authors:** Xiangyu Meng, Xiao Bai, Angting Ke, Kaiqiang Li, Yun Lei, Siqi Ding, Dongqiu Dai

**Affiliations:** 1Department of Surgical Oncology, the Fourth Affiliated Hospital of China Medical University, Shenyang 110032, China; mengxiangyu@cancerhosp-ln-cmu.com (X.M.); xiaobai@cmu.edu.cn (X.B.); kqli@cmu.edu.cn (K.L.); 2021121651@cmu.edu.cn (Y.L.); sqding@cmu.edu.cn (S.D.); 2Cancer Center, the Fourth Affiliated Hospital of China Medical University, Shenyang 110032, China; 3Department of Gastric Surgery, Cancer Hospital of China Medical University, Liaoning Cancer Hospital, Shenyang 110042, China

**Keywords:** gastric cancer, drug resistance-related lncRNAs, mechanisms, clinical application

## Abstract

Gastric cancer (GC) ranks as the third most prevalent malignancy and a leading cause of cancer-related mortality worldwide. However, the majority of patients with GC are diagnosed at an advanced stage, highlighting the urgent need for effective perioperative and postoperative chemotherapy to prevent relapse and metastasis. The current treatment strategies have limited overall efficacy because of intrinsic or acquired drug resistance. Recent evidence suggests that dysregulated long non-coding RNAs (lncRNAs) play a significant role in mediating drug resistance in GC. Therefore, there is an imperative to explore novel molecular mechanisms underlying drug resistance in order to overcome this challenging issue. With advancements in deep transcriptome sequencing technology, lncRNAs—once considered transcriptional noise—have garnered widespread attention as potential regulators of carcinogenesis, including tumor cell proliferation, metastasis, and sensitivity to chemo- or radiotherapy through multiple regulatory mechanisms. In light of these findings, we aim to review the mechanisms by which lncRNAs contribute to drug therapy resistance in GC with the goal of providing new insights and breakthroughs toward overcoming this formidable obstacle.

## 1. Introduction

GC ranks as the third most prevalent malignancy globally and represents a significant cause of cancer-related mortality [1]. A good prognosis of patients with GC benefits from the early diagnosis and treatment including drug and surgical treatments. However, the majority of patients with gastric cancer are typically diagnosed at an advanced stage [1,2]. Perioperative and postoperative chemotherapies play a key role in preventing relapse and metastasis for these patients with middle- and late-stage GC [1]. The overall treatment effect, however, remains efficacious with a mere 20–40% five-year overall survival (OS) rate observed in patients with advanced GC undergoing standard therapy [3]. The main obstacle in GC therapy is intrinsic or acquired chemoresistance [4]. Hence, exploring some novel molecular mechanisms of chemoresistance is urgently needed.

With the improvement in deep transcriptome sequencing technology, lncRNAs, which were initially considered transcriptional noise, have attracted widespread research attention [5]. LncRNAs are longer than 200 nucleotides and exert important roles during the universal biological processes, including cell differentiation, chromatin reorganization and modification, immune responses, and carcinogenesis [6,7]. In particular, lncRNAs significantly contribute to transcription and splicing, as well as organellar biogenesis, subcellular molecular trafficking, and cell differentiation [8,9]. Although lncRNAs are not translated into proteins, they can function as oncogenes or tumor suppressors by upregulating or downregulating the expression of protein-coding genes in cancers [10]. Furthermore, mounting evidence suggests that dysregulated lncRNAs are intricately associated with the development of chemoresistance in various types of cancers [11].

In recent decades, significant advancements have been made in chemotherapy regimens and the management of drug side effects for the treatment of advanced GC, with commonly used drugs such as 5-fluorouracil (5-Fu), cisplatin (DDP), or paclitaxel. Nevertheless, the effectiveness of chemotherapy remains limited, and the 5-year overall survival rate for advanced patients with gastric cancer undergoing this treatment modality still remains low [12]. The main reason leading to a struggle with chemotherapy is still tumor drug resistance in GC [13]. It is known that drug resistance in tumor cells includes primary resistance (existing without exposure to the drug) and acquired resistance (gradual development induced by chemotherapy drugs), which seriously affect the treatment effect on patients with cancer [14].

Drugs can also develop cross-resistance, also known as multidrug resistance (MDR). Tumor cells may evade effective chemotherapy through many different strategies, such as increased anti-apoptotic ability, abnormal deoxyribonucleic acid (DNA) damage repair ability, upregulation of adenosine triphosphate (ATP)-dependent drug transport pump, and changes in drug metabolisms [15,16]. Recent studies have shown that lncRNAs acting as carcinogenic or tumor suppressor genes can increase or decrease the fight against tumor sensitivity to drugs. To further explore the mechanism of lncRNAs related to chemotherapy resistance in GC, we reviewed the mechanisms of lncRNA-related drug resistance to provide new ideas and breakthroughs for solving the difficult problem of drug resistance in GC.

## 2. Overview of lncRNAs

RNA is between DNA and protein and acts as a bridge during the processes driving genetic information and biological functions induced by most proteins. Studies have shown that about 98% of the DNA sequence in the whole genome is a non-coding sequence, but less than 2% of the genes encode proteins [17]. These lncRNAs were previously regarded as “transcriptional noise” in vivo. With the continuous development and updates in sequencing technology in recent years, researchers have gradually paid attention to lncRNAs and found that they have special functions in the vast majority of biological processes in the body [18,19].

According to different sequence lengths, ncRNA includes the following two categories: small ncRNA and lncRNA. LncRNAs are non-coding RNAs that are longer than 200 nt [20,21,22]. Although there are some ncRNAs that encode protein functions, most of them do not encode proteins [23]. So far it is believed that lncRNAs are mainly derived from the following pathways: (1) the frame of the protein-coding gene is disrupted and is transformed into a functional non-coding RNA that incorporates some previous coding sequence; (2) following chromosomal rearrangement, two previously distant and transcriptionally inactive sequence regions become juxtaposed, resulting in the formation of a multi-exonic non-coding RNA; (3) the duplication of a non-coding gene through retro-transposition results in the formation of either a functional non-coding retrogene or a nonfunctional non-coding retro-pseudogene; (4) adjacent repetitive sequences in non-coding RNA originate from two tandem duplication events; and (5) the insertion of a transposable element form the functional non-coding RNA [18]. LncRNAs have a stable structure and function because of their short flanking sequences. Although they are numerous in quantity, they are poorly conserved [19,24,25,26,27].

LncRNAs consist of nuclear lncRNAs and cytoplasmic lncRNAs according to their cellular localization. Nuclear lncRNAs are responsible for modulating chromatin structure and function, regulating the transcription of neighboring and distant genes, and affecting RNA splicing, stability, and translation [28]. Cytoplasmic lncRNAs can maintain the stability and transcription efficiency of their target mRNA through RNA–protein interactions [29]. In addition, some functions of cytoplasmic lncRNAs are also associated with ribosomes, mitochondria, and other organelles including exosomes [28,29]. Based on their genomic location relative to protein-coding genes, lncRNAs can be categorized into different groups as follows: intergenic lncRNA, intronic lncRNA, divergent lncRNA, antisense lncRNA, and enhancer lncRNA [26,30] (Figure 1).

## 3. Drug Resistance-Related lncRNAs and Malignant Phenotypes in GC

It is widely acknowledged that lncRNAs with “genetic noise” characteristics play crucial roles in the initiation and progression of GC. By affecting genetic modification and transcriptional regulation, lncRNAs relate to multiple biological processes in GC including proliferation, metastasis, apoptosis, and drug resistance. Especially in drug resistance regulation, lncRNAs often induce changes in drug-resistant cells in the malignant phenotype with the help of the abnormal expression of lncRNAs, which eventually leads to drug resistance and disease progression in patients with GC after drug treatment (Table 1).

### 3.1. Drug Resistance-Related lncRNAs and Metastasis

Cancer metastasis occurs frequently in solid tumors, namely, because of the spread of a tumor to distant parts of the body from its original site, which remains the cause of 90% of deaths from solid tumors [102]. The complex processes enable tumor cells to achieve metastasis. A considerable amount of evidence indicates that ncRNAs including lncRNAs may participate in cancer metastasis [103,104]. Particularly in drug-resistant tumors, the alteration in the tumor microenvironment (TME) induced by various mechanisms of drug resistance facilitates interactions among tumor cells, stromal cells, and the secretion of soluble inflammatory molecules. These interactions mediate the recruitment of immune cells that enhance tumor cell survival and promote metastasis [105]. Therefore, lncRNAs associated with drug resistance play an outstanding functional role in regulating metastasis in GC.

In DDP-resistant GC cell lines, 12 lncRNAs were found to be involved in the regulation and promotion of invasion and metastasis. They are HOTAIR [60,61,62], HOTTIP [65], FAM84B-AS [48], KLF3-AS1 [69], SNHG6 [93], ST7-AS1 [106], LINC-PINT [72], and LINC00922 [73]. Six lncRNAs are implicated in the multidrug resistance of gastric cancer cells and facilitate the invasion and metastasis of these drug-resistant cells. They are PVT1 [88,89,90], ANRIL [34,35], MALAT1 [76,77,78,79,80,81], ZFAS1 [101], SUMO1P3 [95], D63785 [46], and SNHG1 [107], which mediate the invasion and metastasis of taxane-resistant GC cell lines through the upregulation of their own expression. LEIGC [70] and HNF1A-AS1 [59] are also vital in promoting the invasion and metastasis of fluorouracil-resistant GC cell lines. This literature review revealed the involvement of three distinct lncRNAs in the mechanism underlying invasion and metastasis in platinum-resistant gastric cancer cell lines. These are BLACAT1 [39], DUSP5P1 [45], and SLCO1C1 [91]. PD1 monoclonal antibody treatment is a new tumor immunotherapy method checkpoint blockade therapy. Anti-PD-1 monoclonal antibody is a new method for tumor immunotherapy [108,109]. The efficacy of anti-PD-1 antibody monotherapy in advanced GC has been confirmed and supported by several trials [110,111]. In this literature review, lncRNAs were also found to be involved in the regulation of invasion and metastasis of PD-1-resistant GC cell lines, such as NUTM2A-AS1 [85]. In addition, the invasion and metastasis of some drug-resistant cell lines that are uncommon in the treatment of GC are regulated by lncRNAs. For example, NEAT1 can promote the invasion and metastasis of Adriamycin-resistant cell lines [84], and MVIH plays a crucial part in positively regulating the invasion and metastasis of GC gemcitabine-resistant cell lines [83].

The majority of studies have suggested that a distinct subpopulation of tumor cells possessing stem-like properties actively participates in the initiation, invasive growth, and metastasis of tumors [112]. It should be noted that there are still some lncRNAs that alter the TME by regulating cancer stem cell activity and then promoting the invasion and metastasis of drug-resistant GC cell lines. For example, BCAR4 [38], THOR [96], and LINC00942 [74] participate in the positive regulation of invasion and metastasis of GC platinum-resistant cell lines by promoting the activation of tumor stem cells. Similarly, the pivotal involvement of MALAT1 [76,77,78,79,80,81], MACC1-AS1 [75], and HCP5 [55,56,57] in orchestrating the invasion and metastasis of multidrug-resistant cell lines in GC cannot be overstated.

The epithelial-to-mesenchymal transition (EMT) has been acknowledged as a pivotal characteristic of tumor metastasis [113,114]. The process of EMT facilitates the detachment of tumor cells from the primary mass by diminishing their cell adhesive properties, ultimately promoting local invasion, intravasation into blood or lymph vessels, extravasation, and subsequent re-establishment of the primary mass at distant sites [113]. Based on the redifferentiation properties of cancer stem cells themselves, researchers have found that the metastatic outcome of EMT may be caused by the colonization and redifferentiation of tumor stem cells in distant organs [112,115]. In our review, some lncRNAs were also observed to be involved in mediating the EMT of GC drug-resistant cell lines to cause invasion and metastasis. For example, LEIGC [70], ZFAS1 [101], HOTAIR [63], MALAT1 [78,79], MVIH [83], H19 [68], KLF3-AS1 [69], SNHG6 [93], HNF1A-AS1 [59], and ST7-AS1 [106] (Figure 1).

### 3.2. Drug Resistance-Related lncRNAs and Proliferation

Malignancy is considered a disease in which proliferation is a loss of management. Uncontrolled proliferation assumes a pivotal role in the progression of tumorigenesis [116]. The abnormal regulation of mitosis, proliferation signal-related gene mutations, and other abnormalities of proliferation signaling pathways eventually lead to a significantly shortened cell cycle and accelerate abnormal cell proliferation [117]. In this review, we also found that many lncRNAs were involved in regulating the proliferation of multiple drug-resistant GC cell lines. For example, in DDP-resistant cell lines, 14 lncRNAs were involved in regulating the proliferation in GC cell lines. These lncRNAs are HOTAIR [60,61,62,63], HULC [66,67], HOTTIP [65], DANCR [44], FAM84B-AS [48], HMGA1P4 [58], ASB16-AS1 [37], KLF3-AS1 [69], SNHG6 [93], FOXD1-AS1 [37], PITPNA-AS1 [87], ST7-AS1 [106], LINC-PINT [72], an LINC00922 [73]. In 5-FU-, oxaliplatin-, and docetaxel-resistant cell lines, there were five cases (LEIGC [70], MACC1-AS1 [75], FGD5-AS1 [50], HAGLR [54], and HNF1A-AS1 [59]), and two cases (lncRNA), respectively, (DUSP5P1 [45] and SLCO1C1 [91]), and two lncRNAs (MRUL [82], D63785 [46]) were involved in regulating the proliferation of these drug-resistant GC cell lines. In addition, 11 lncRNAs have been found to regulate multidrug resistance of GC, these lncRNAs are PVT1 [88,89,90], UCA1 [97,98,99,100], ANRIL [34,35], MALAT1 [76,78,79,80,81,118], PANDAR [86], ZFAS1 [101], ARHGAP5-AS [36], HCP5 [55,56,57], FEZF1-AS1 [49], SNHG12 [94], and SUMO1P3 [95]. In Adriamycin-resistant GC cells, NEAT1 can promote cell proliferation [84], and MVIH can also promote the proliferation of gemcitabine-resistant GC cell lines [83]. For the targeted therapy of GC, this review also found that two lncRNAs were involved in promoting the proliferation of drug-resistant cell lines. Among them, NUTM2A-AS1 can increase the proliferation ability of PD-1-resistant GC cell lines [85]. CRART16 can accelerate the proliferation of bevacizumab-resistant GC cell lines [41]. It is worth noting that some lncRNAs, such as GAS5 [53] and ADAMTS9-AS2 [32], were found to have inhibitory effects on the proliferation of drug-resistant gastric cancer cell lines in this review. Numerous articles have shown that GAS5 can control the proliferation of Adriamycin-resistant gastric cancer cell lines [53]. ADAMTS9-AS2 can restrain the proliferation of platinum-resistant GC cell lines [32].

Similarly, some lncRNAs also indirectly promote the proliferation of drug-resistant gastric cancer cell lines by regulating cancer stem cell properties. For example, in a platinum-resistant GC cell line, these lncRNAs include BCAR4 [38], THOR [96], MACC1-AS17 [75], HCP5 [55,56,57], ASB16-AS1 [37], and LINC00942 [74] (Figure 1).

### 3.3. Drug Resistance-Related lncRNAs and Apoptosis

The dysregulation of apoptosis in tumors represents a pivotal hallmark of cancer. The aberrant regulation of multiple apoptotic signaling pathways is implicated in the sustained survival of tumor cells following detachment from apoptosis. Apoptosis is regulated through the following two principal pathways: the intrinsic or mitochondrial pathway, predominantly activated by intracellular stress signals, and the extrinsic or death receptor pathway, triggered by extracellular signals [119]. Research shows that among the various mechanisms that facilitate MDR, disordered apoptosis and programmed cell death have disastrous consequences for drug therapy [119]. At the same time, a variety of apoptosis-related proteins in the apoptotic signaling pathway closely take part in the regulation of GC drug resistance such as Bcl-2 [120], inhibitors of apoptosis family members [121], and p53 [122]. The dysregulation of apoptosis is a crucial phenomenon that confers drug resistance to cancer cells [123]. LncRNAs have a pivotal role in governing the apoptosis of drug-resistant GC cell lines.

In this review, we found that the mechanism by which most lncRNAs regulate apoptosis in drug-resistant GC cell lines is mainly concentrated in the intrinsic or mitochondrial pathway. MRUL [82], AK022798 [33], PVT1 [88,89,90], UCA1 [97,98,99,100], CASC9 [124], HOTAIR [61,62,63,64], BLACAT1 [39], SNHG5 [92], FAM84B-AS [48], SNHG6 [93], APAF1 [31] and SNHG8 [74] have demonstrated the ability to downregulate protein levels of caspase 3, caspase 7, caspase 8, and caspase 9, thereby inhibiting cleaved PARP protein expression while promoting the expression of the anti-apoptotic protein Bcl-2. These events eventually make the corresponding drug-resistant GC cell lines show a high survival state and induce drug resistance. Moreover, certain lncRNAs have been identified to exert their influence on drug-resistant GC cell lines by modulating DNA damage repair pathways, including lncRNA CRAL, thereby impeding apoptosis [125].

In addition, in platinum-resistant GC cell lines, HULC [66,67], HOTTIP [65], DANCR [44], HMGA1P4 [58], PITPNA-AS1 [37], ST7-AS1 [106], LINC00922 [88], and LINC00942 [74] could inhibit the apoptosis of drug-insensitive GC cell lines. In multidrug-resistant GC cell lines, some lncRNAs can also inhibit apoptosis from occurring, such as ANRIL [34,35], MALAT1 [76,77,78,79,80,81], PANDAR [86], ZFAS1 [101], ARHGAP5-AS [36], HCP5 [55,56,57], EIF3J-DT [47], CRNDE [42,43], and ABL [31]. In Adriamycin-resistant cell lines, NEAT1 [84] and H19 have an inhibitory effect on cell apoptosis [68]. In gemcitabine-resistant cell lines, lncRNA MVIH [83] can also play an inhibitory role in cell line apoptosis. On the contrary, some lncRNAs also can make GC cells more sensitive to drugs and reduce their apoptotic effects, such as GAS5 [53] and ADAMTS9-AS2 [32].

There are still some lncRNAs that affect the apoptosis function of drug-resistant GC cell lines through some special regulatory ways as follows: (1) By regulating the stemness of GC cells: These lncRNAs include BCAR4 [38], THOR [96], MACC1-AS1 [75], MALAT1 [78], and ASB16-AS1 [37]. These lncRNAs can regulate the expression of cancer stem cell markers including β-catenin, c-Myc, and Klf4 to induce and enhance the stem-like characteristics of drug-resistant GC cell lines and inhibit apoptosis to regulate the drug resistance. (2) By regulating the autophagy of GC cells: Autophagy is a multi-stage conservative dysfunctional process that contributes to maintaining cellular homeostasis. Aberrant autophagy is intricately associated with the onset, progression, and acquisition of drug resistance in tumors. During the action of autophagy, tumor cells are more able to tolerate external stress stimulation such as hypoxia and starvation and enhance the proliferation and prolong the survival of cancer cells [126]. Autophagy is closely related to GC resistance. On the one hand, autophagy can facilitate chemoresistance and prolong GC cell life, whereas in others, it can also promote chemosensitivity and accelerate cell death [127]. Multiple factors are involved in the regulation of autophagy, such as AMPK, MAPK, PI3K-AKT, BECN1, and ATG proteins. Pro-survival autophagy can ensure that cancer cells are protected as much as possible from the toxic and damaging effects of chemotherapy drugs. Considering the extensive regulatory role of lncRNAs in tumorigenesis and development, it is noteworthy that lncRNAs also exert an influence on autophagy function in drug-resistant GC cell lines. For example, MALAT1 facilitates drug resistance by modulating the expression of autophagy-related proteins, thereby inducing autophagy and subsequently suppressing apoptosis in drug-resistant GC cell lines [76,77,80]. In platinum-resistant GC cell lines, HULC promotes autophagy by managing *FoxM1* expression, thereby inhibiting apoptosis in drug-resistant cell lines [67]. In oxaliplatin and fluorouracil-resistant GC cell lines, EIF3J-DT promotes autophagy by increasing the expression of ATG14 protein, which eventually leads to a reduction in apoptosis in the two dru- resistant cell lines and mediates drug resistance [47]. Similar regulatory mechanisms have also been found in the regulation of autophagy by lncRNA 01572 [71] and FEZF1-AS1 [49]. In addition to inducing autophagy, some lncRNAs can also regulate drug sensitivity through autophagy in drug-resistant cell lines. For example, CRNDE [43] and PINT [72]. (3) By regulating ferroptosis of tumor cells: The process of ferroptosis involves iron-dependent lipid peroxidation, leading to cell death. The discovery of ferroptosis significantly expanded the repertoire of cell death modalities, offering novel insights for further investigation into the biological behaviors of gastric cancer, including proliferation, metastasis, invasion, and drug resistance. Iron-dependent lipid peroxidation is an important marker of ferroptosis [128]. Many studies reveal that the drug resistance of GC cells can be modulated through ferroptosis metabolism. The study conducted by Xiao et al. demonstrated that the utilization of a model based on differentially expressed lncRNAs associated with ferroptosis exhibited significant predictive efficacy in determining the response to drug treatment for GC [129]. In platinum-resistant GC cell lines, CBSLR can inhibit ferroptosis under the induction of a hypoxia environment, contributing to the resistance of GC cells [40].

It can be seen that lncRNAs are often involved in the regulation of the apoptosis of drug-resistant GC cells in a multi-pathway and complex manner, which also indicates that lncRNAs have a very important position in the effect on the apoptosis of drug-resistant GC cells (Figure 1).

## 4. Mechanisms of Drug Resistance-Related lncRNAs in GC

Previous studies have demonstrated the extensive involvement of lncRNAs in the regulation of chemoresistance mechanisms in gastric cancer. The underlying mechanisms by which lncRNAs contribute to drug resistance in GC primarily encompass oncogene amplification and overexpression, anti-apoptosis, immune escape, epigenetic modification, upregulation of multiple drug resistance (MDR)-related genes, and so on [130,131,132].

### 4.1. The lncRNA-miRNA(-mRNA) Network

#### 4.1.1. ceRNA Network

Since the concept of “ceRNA” was proposed in 2011, ceRNA network regulation has acted as an important function in gene post-transcriptional regulation. ncRNAs containing miRNA response elements (MREs) compete with miRNA target gene-mRNA to bind miRNAs, eventually leading to the weakening or disappearance of the inhibition of downstream target genes and the release of the inhibition of target mRNA expression [132]. The ceRNA network regulation mechanism contains ncRNAs (lncRNAs/circRNAs/pseudogenicRNAs), microRNAs, and mRNAs. In the regulatory mechanism of drug resistance in GC, this review also identified that many lncRNAs can form the ceRNA network with other ncRNAs, which is a vital mechanism of drug resistance regulation in GC [132] (Table 2).

As shown in Table 2, a total of 33 lncRNAs regulate drug resistance in GC cell lines through the ceRNA management mechanism. LncRNAs induce the upregulation of downstream target genes, which eventually leads to different regulatory mechanisms and then induces drug resistance of GC. The ceRNA regulatory network eventually leads to different pathological processes in drug-resistant cell lines, mainly including upregulating the expression of target genes, regulating the signaling pathway, increasing activity of transcription factors, inducing autophagy, regulating epigenetic modification, increasing metabolic regulation, promoting metastasis and angiogenesis, reducing the immune response, and regulating proliferation. We also found that the lncRNA-miRNA-mRNA interaction is not one-to-one binding but one-to-many binding, for example, HOTAIR [61,63,64], MALAT1 [76,77,80,81], NUTM2A-AS1 [85], ASB16-AS1 [37], and SLCO1C1 [91]. Therefore, this ceRNA regulatory network is not a single regulation but a complex multi-path regulation. This result is also consistent with Arancio W et al. [133]. It is worth noting that there are also some lncRNAs that increase the sensitivity of GC chemotherapy drugs through the ceRNA regulation mechanism, such as CRAL [125]. Wang et al. showed that lncRNA CRAL could increase the sensitivity of GC cell lines to DDP by adsorbing miR-505 to promote *CYLD* gene expression and inhibit AKT signaling pathway activation. The details are shown in Table 2.
biomolecules-14-00608-t002_Table 2Table 2The ceRNA network formed by drug resistance-related lncRNAs and their function in gastric cancer.LncRNAsExpressionSponging miRNAsTargetsFunctionsDrugsRefs.Upregulating the expression of target genes



BLACAT1↑miR-361*ABCB1*upregulating *ABCB1* expressionOXA[39]CRART16↑miR-122-5p*FOS*upregulating *VEGFD* expressionBevacizumab[41]FENDRR↑miR-4700-3p*FOXC2*upregulating *FOXC2* expressionVCR, ADM[131]HIF1A-AS2↑miR-29c*LOX*upregulating *LOX* expressionMDR[134]HOTAIR↑miR-195-5p*ABCG2*upregulating *ABCG2* expressionOXA[64]LINC00922↑miR-874-3p*GDPD5*upregulating *GDPD5* expressionDDP[73]MALAT1↑miR-22-3p*ZFP91*upregulating *ZFP91* expressionOXA[78]PCAT-1↑miR-128*ZEB1*upregulating *ZEB1* expressionDDP[130]PVT1↑miR-3619-5p*TBL1XR1*upregulating *TBL1XR1* expressionDDP[88]Activating signaling pathways



ASB16-AS1↑miR-3918, miR-4676-3p*TRIM37*activating NF-kappa B pathwayDDP[37]CRAL↓miR-505*CYLD*suppressed AKT activationDDP[125]D63785↑miR-422a*MEF2D*activating VEGF/TGF-β1 pathwayDOX[46]FOXD1-AS1↑miR-466*PIK3CA*activating PI3K/AKT/mTOR pathwayDDP[52]HCP5↑miR-3619-5p*PPARGC1A*activating AMPK pathway5-FU, OXA[96]HOTAIR↑miR-126*VEGFA, PIK3R2*activating PI3K/AKT/MRP1 pathwayDDP[61]SNHG17↑miR-23b-3p*Notch2*activating Notch2 pathwayDDP[135]TMEM44-AS1↑miR-2355-5p*PPP1R13L*inhibiting p53 pathway5-FU[136]Regulating the activity of transcription factors



EIF3J-DT↑miR-188-3p*ATG14*inducing autophagy5-FU, OXA[92]FGD5-AS1↑miR-153-3p*CITED2*increasing transactivator activity5-FU[50]HCP5↑miR-128*HMGA2*increasing transcriptional activityDDP[56]
↑miR-519d*HMGA1*increasing transcriptional activityDDP[57]HOTTIP↑miR-218*HMGA1*upregulating transcriptional regulator expressionDDP[65]Inducing autophagy



LINC01572↑miR-497-5p*ATG14*inducing autophagyDDP[71]MALAT1↑miR-23b-3p*ATG12*inducing autophagyDDP, VCR, 5-FU[76]
↑miR-30b*ATG5*inducing autophagyDDP[77]
↑miR-30e*ATG5*inducing autophagyDDP[80]Epigenetic modification



CRAL↓miR-505*CYLD*suppressing AKT activationDDP[125]HOTAIR↑miR-17-5p*PTEN*inhibiting the PTEN phosphatase activityDDP, ADM, MMC, 5-FU[63]Metabolic regulation



HAGLR↑miR-338-3p*LDHA*increasing glycolysis5-FU[54]HCP5↑miR-3619-5p*PPARGC1A*increasing fatty acid oxidation5-FU, OXA[55]SNHG1↑miR-216b-5p*HK2*increasing glycolysisPTX[107]SNHG7↑miR-34a*LDHA*increasing glycolysisDDP[137]SNHG16↑miR-506-3p*PTBP1*increasing glycolysis5-FU[138]Metastasis and angiogenesis



CRART16↑miR-122-5p*FOS*increasing angiogenesisBevacizumab[41]HNF1A-AS1↑miR-30b-5p*EIF5A2*promoting EIF5A2-induced EMT process5-FU[59]Apoptosis regulation



ADAMTS9-AS2↑miR-223-3p*NLRP3*activating NLRP3 mediated pyroptotic cell deathDDP[32]SNHG6↑miR-1297*Bcl-2*inhibiting apoptosisDDP[93]UCA1↑miR-513a-3p*CYP1B1*inhibiting apoptosisDDP[100]Immune response regulation



NUTM2A-AS1↑miR-376a*TET1, HIF-1A*inhibiting immune responsesPD-L1[85]SNHG15↑miR-141*PD-L1*inhibiting immune responsesPD-L1[139]Proliferation regulation



ANRIL↑miR-181a-5p*CCNG1*inhibiting proliferationDDP[35]SLCO1C1↑miR-204-5p, miR-211-5p*SSRP1*enhancing cell growth, preventing DNA damageOXA[91]Abbreviations: DDP: cisplatin; ADM: Adriamycin; MMC: mitomycin; 5-Fu: fluorouracil; VCR: vincristine; OXA: oxaliplatin; MDR: multidrug resistance; ↑: high expression; ↓: low expression.


#### 4.1.2. lncRNA-miRNA

There are also some lncRNAs that regulate the drug resistance of GC by adsorbing miRNA. This lncRNA-miRNA interaction mechanism is essentially derived from the mechanism in which lncRNAs containing MREs bind to miRNAs. For example, the overexpression of lncRNA CASC2 increased the resistance effect for DDP in GC by sponging miR-19a [140]. He et al. demonstrated that mesenchymal stem cells (MSCs) secrete TGF-β1, which induces the upregulation of MACC1-AS1 at the RNA level. This upregulation activates fatty acid oxidation-dependent stemness and chemoresistance by antagonizing miR-145-5p in patients with gastric cancer undergoing FOLFOX treatment [140]. LncRNA KLF3-AS1 enhanced chemosensitivity to cisplatin by inhibiting miR-223 expression [69]. The exosome-mediated transfer of FGD5-AS1 disseminates the DDP resistance effect among GC cells by absorbing miR-195 [51]. Although these lncRNA-miRNA interactions do not find downstream target mRNAs, based on the complex information regulatory network in organisms, this does not mean that lncRNA-miRNA interactions exist alone and do not affect downstream factors or pathways.

### 4.2. Upstream Regulation of Drug Resistance-Related lncRNAs in GC

In addition to the spatial expression limitation of lncRNA expression, a variety of molecules can affect the level of lncRNA through the regulation of the lncRNA gene promoter [19]. These [141] molecules include signaling pathway receptors, cytokines, bio-enzymes, modified proteins, transcription factors, and exogenous molecules.

For instance, Notch 1 can increase the transcription of gene targets by interacting with Notch ligands. Notch 1 protein binds to lncRNA AK022798 and increases the expression of AK022798, which promotes cisplatin-resistant gastric cancer formation, resulting in the upregulating expression of MRP1 and P-glycoprotein and reduces the apoptosis of DDP-resistant cells [33]. Cytokines, such as TGF-β1 and midkine, play a crucial role in the regulation of lncRNA expression and the induction of drug resistance. TGF-β1 derived from MSCs activates SMAD2/3 signaling by binding to TGF-β receptors, thereby upregulating the expression of lncRNA MACC1-AS1 in GC cells. This activation leads to fatty acid oxidation-dependent stemness and chemoresistance by antagonizing miR-145-5p [75]. Yang et al. demonstrated that cancer-associated fibroblast (CAF)-derived midkine (MK) could induce the upregulation of long non-coding RNA ST7-AS1 in DDP-resistant GC cells, resulting in enhanced phosphorylation of PI3K and AKT and subsequent activation of the PI3K/AKT pathway, thereby facilitating resistance to DDP [106]. Furthermore, certain enzymes have been identified to facilitate aberrant lncRNA expression. For instance, methioninase (METase), also known as l-methionine-α-amino-γ- mercaptoethane lyase, has been documented to play significant roles in suppressing cancer growth and overcoming drug resistance [135]. In DDP-resistant gastric cancer cells, overexpressed METase could downregulate the expression of lncRNA HULC, thus decreasing the protein level of FoxM1 and suppressing autophagy and cisplatin resistance [67]. Also in DDP-resistant GC cell lines, HDAC3, as a class of histone deacetylase, plays important roles in epigenetic regulation and gene transcription [142]. A study by Ren et al. suggested that HDAC3 promoted the transcription of lncRNA LOC101928316 by decreasing the level of acetylation of H3K4 on its promoter, resulting in GC cell resistance to cisplatin by promoting the PI3K-Akt-mTOR pathway [143]. In addition, some modifier proteins can also modify the upstream transcription process of lncRNA to affect the expression of lncRNA. SUMO1 protein was found to inhibit the SUMOylation level of SP1, which led to upregulated lncRNA SNHG17 expression in the SNHG17 promoter. The high level of lncRNA SNHG17 can bind to miR-23b-3p as a sponge, leading to decreased inhibition of the target gene *Notch2*, which finally promotes the resistance of GC to DDP [144]. As a member of RNA-binding proteins, transcription factors were also closely related to the regulation of lncRNA expression. E2F6, one of the known E2F transcription factor families, has a classical transcriptional inhibitor function by downregulating the transcription of downstream genes [145]. A study by Zhang et al. determined that E2F6 could decrease the expression of CRNDE by binding to the CRNDE promoter at nucleotides 660–670, thereby promoting autophagy and inhibiting apoptosis [42]. In this review, we also found that some exogenous molecules are involved in the regulation of lncRNA expression. For example, Ma et al. showed that dioscin, an active ingredient identified in edible medicinal plants, might block the cell cycle of GC by downregulating the expression level of HOTAIR [60].

So, it follows that lncRNA can bind to a variety of molecules, including proteins. It is regulated by epigenetic modification, the transcriptional level, and other mechanisms, which eventually leads to changes in lncRNA expression and induced drug resistance (Figure 2A).

### 4.3. Downstream Regulation of Drug Resistance-Related lncRNAs in GC

The function of lncRNAs is intricately linked to their subcellular localization [146]. Nuclear lncRNAs are involved in nuclear processes such as chromatin organization, RNA transcription, and splicing. Cytoplasmic lncRNAs regulate mRNA transport, as well as protein stability, and posttranslational modification [26]. In this review, some drug resistance-related lncRNAs were found to play different roles in mediating drug resistance because of their different spatial locations. LncRNAs related to drug resistance located in the nucleus can exert scaffolding or enhancer functions to affect the expression of downstream genes or proteins. Cytoplasmic lncRNAs can act as signal transduction or post-transcriptional modifiers to affect the expression of downstream genes or proteins.

#### 4.3.1. Scaffold-like lncRNAs in the Nucleus

For instance, PANDAR exerts regulatory control over the transcription of the *CDKN1A* gene by competitively binding with the p53 protein at the CDKN1A promoter in a p53-dependent manner, thereby augmenting the resistance of gastric cancer cell lines to oxaliplatin [86]. The interaction between lncRNA HULC and FoxM1 contributes to the enhancement of cisplatin resistance in drug-resistant GC cells by regulating *FoxM1* gene expression and facilitating autophagy [67]. Enhanced by zeste homolog 2 (EZH2), a member of the polycomb group genes (PcGs) family, exerts epigenetic modifications on gene expression through transcriptional repression [147]. Many EZH2-related drug-resistant lncRNAs in the nucleus have scaffolding functions that affect gene or protein expression. UCA1 inhibits cisplatin-induced apoptosis by interacting with EZH2 and regulating *EZH2* expression, thus activating the PI3K/AKT pathway [99]. PCAT1 decreases PTEN by binding to EZH2 at the *PTEN* promoter, thus increasing H3K27me3 and promoting cisplatin resistance in GC cells [148]. LINC-PINT attracted an enhancer of *EZH2* at the promotor of *ATG5* to downregulate the transcription level of *ATG5*, leading to the suppression of autophagy and DDP resensitization [72]. Other drug resistance-associated lncRNAs also function as scaffolding and play important roles in regulating gene expression. For example, CRNDE functions as a scaffold to recruit NEDD4-1 to PTEN, thereby facilitating the NEDD4-1-mediated degradation of PTEN and subsequently reducing its protein levels. This molecular mechanism significantly impacts cisplatin resistance in gastric cancer [149]. In addition, CRNDE also functions as a proteasome and could interact with SRSF6. CRNDE can degrade the SRSF6 protein through ubiquitination and then affect the cleavage of SRSF6 in the nucleus, ultimately promoting autophagy and regulating chemotherapy resistance [43]. SNHG8 was found to interact with hnRNPA1, an RNA binding protein, enhanced the stability of the TROY protein, and regulated the level of the TROY protein in GC cell lines [150] (Figure 2B).

#### 4.3.2. Enhancer-like lncRNAs in the Nucleus

As a small region of DNA, enhancers can bind to the protein, which strengthens the transcription of a gene. Some lncRNAs associated with drug resistance also bind with enhancers to enhance gene transcription. For instance, MURL has an enhancer-like role by inducing *ABCB1* transcription in MDR gastric cancer cells [82]. THOR was found to be directly bound to the 3′ UTR region of the *SOX9* gene and thus increased SOX9 expression and mRNA stability in the DDP drug resistance of GC cells [96]. Studies have shown that in the stemness regulation of GC, lncRNA MALTAT1 can regulate the expression of dry marker protein family genes upstream [78]. DANCR accelerated the multidrug resistance of GC by upregulating the expressions of MDR1 and MRP1 [44]. FEZF1-AS1 was found to be closely related to autophagy and could regulate the chemo-resistance of GC cells by directly targeting *ATG5* and increasing the expression of ATG5 [49]. SNHG12 was found to be bound to HuR and compose a “SNHG12-HuR” complex in the cytoplasm. Then, the complex increased the stability of YWHAZ mRNA, thus promoting GC cell proliferation and chemoresistance [94]. HIT000218960 was found to activate AKT/mTOR/P70S6 kinase (P70S6K) by regulating *HMGA2* expression, thus enhancing resistance to 5-Fu in GC cells [151]. SUMO1P3 could directly interact with CNBP, which activated its downstream oncogenes such as *c-myc* and *cyclin D1*, and promoted drug resistance in GC [95]. DUSP5P1 promoted *ARHGAP5* transcription by gathering with the promoter of *ARHGAP5* and focal adhesion and MAPK pathway, which promoted metastasis and platinum drug resistance in GC [45] (Figure 2C).

#### 4.3.3. Transcription and Translation Functions of lncRNAs in the Cytoplasm

Although lncRNAs mainly function in the nucleus, cytoplasmic lncRNAs also perform some special roles in mRNA spatial regulation, transcription, and translation. But mainly, most lncRNAs related to drug resistance are transmitted from the nucleus to the cytoplasm to stabilize some important proteins in the cytoplasm. LncRNAs associated with drug resistance that function similarly include the following: ARHGAP5-AS1 [36], SNHG12 [94], OVAAL [152], and CBSLR [40]. By recruiting METTL3, ARHGAP5-AS1 stabilizes ARHGAP5 mRNA in the cytoplasm, thereby stimulating m6A modification of ARHGAP5 mRNA and upregulating the expression of the ARHGAP5 protein in drug-resistant cells [36]. SNHG12 can bind to HuR and form a “SNHG12-HuR” complex in the cytoplasm, thus increasing the relative HuR expression at the RNA and protein levels and enhancing the stability of YWHAZ mRNA, which promotes GC cell proliferation and chemoresistance [94]. In the cytoplasm, OVAAL can interact with PC (pyruvate carboxylase) and inhibit the ubiquitination of PC mediated by the complex formed by HSC70 and CHIP protein, thereby stabilizing the level of PC protein, which decreases sensitivity to 5-FU treatment [152]. CBSLR combines with YTHDF2 to form a compound that destroys the stability of CBS mRNA by increasing the binding of YTHDF2 with the m6A-modified coding sequence (CDS) of CBS mRNA, which keeps GC cells away from ferroptosis and contributes to chem-resistance in GC [40] (Figure 2B).

### 4.4. Epigenetics and Drug Resistance-Related lncRNAs

As an important epigenetic factor, lncRNAs also play a vital role in epigenetic processes. For example, some drug resistance-related lncRNAs can be found to participate in biological processes such as gene methylation, histone modification, DNA modification, DNA damage repair, and protein ubiquitination. (1) DNA methylation: GAS5 has been shown to promote gene methylation. In ADM-resistant cell lines, GAS5 can sensitize GC cells to Adriamycin (ADM) by increasing the promoter hypermethylation of downstream genes [53]. It is well known that the modification of histones regulates many critical biological processes, usually via chromatin modification that promotes the upregulation or downregulation of target genes [153]. (2) Histone modification: Many lncRNAs are related to the drug resistance of GC-regulated histone modification. For instance, it is well known that EZH2 can act as a histone methyltransferase, which epigenetically inhibits gene expression by increasing H3K27me3 [154,155]. EZH2 can act as a epigenetic regulator together with many lncRNAs, which finally mediate drug resistance in GC. In GC cisplatin-resistant cells, PCAT-1 epigenetically silences *PTEN* by binding to EZH2, thus increasing H3K27me3 and cisplatin resistance in GC [150]. LINC-PINT recruits the EZH2 protein to the promotor of *ATG5*, epigenetically increasing the levels of H3K27me3 to restrict its transcription, leading to the suppression of autophagy and DDP resensitization [72]. Histone acetylation is a typical epigenetic way [156]. Histone acetylation often regulates lncRNA-mediated drug resistance in GC. HDAC3 inhibits the LOC101928316 promoter H3K4ac level to suppress lncRNA-LOC101928316 transcription and activates the PI3K-Akt-mTOR signaling pathway to promote cell activity, invasion, migration, and apoptosis of cisplatin-resistant cell lines [143]. (3) RNA m6A modification: m6A methylation, the most prevalent modification found in both mRNAs and non-coding RNAs, exerts widespread effects on RNA stability, splicing, localization, and translation. In GC multidrug-resistant cell lines, METTL3 can increase the m6A levels of ABL, which interact with IGF2BP1 (a distinct family of m6A readers) and protect APAF1 from forming an apoptotic body, thus inhibiting apoptosis and promoting drug resistance [31]. In a hypoxic tumor microenvironment, CBSLR exerts its potential to promote the survival of GC cells by inhibiting ferroptosis. Mechanistically, CBSLR interacts with YTHDF2 to form a CBSLR/YTHDF2 complex that promotes the destabilization of CBS mRNA by facilitating the enhanced binding of YTHDF2 to the m6A-modified coding sequence (CDS) region of CBS mRNA. Low CBS expression reduces methylation and subsequent ubiquitination degradation of ACSL4 protein [40]. (4) DNA repair: The DNA repair process plays a vital role in carcinogenesis, and its aberrant regulation is responsible for genomic instability [157]. In OXA-resistant cell lines, SLCO1C1 prevents DNA damage. The SLCO1C1 protein acts as a structural support for the SSRP1/H2A/H2B complex, thereby regulating the functional role of SSRP1 in DNA damage inhibition and consequently enhancing oxaliplatin resistance in GC cell lines [91]. SNHG8 in the cytoplasm interacts with hnRNPA1 and enhances its stability by binding to HNRNPA1, thus increasing the level of TROY and damaging DNA damage repair in gastric cancer cell lines during chemotherapy [150]. (5) Protein ubiquitination: As an important component of gene post-transcriptional modification, ubiquitination plays a core function in protein degradation. In the process of the ubiquitin–proteasome system, proteins located in cells are destabilized and degraded through a cascade mediated by ubiquitin initiation [158]. Numerous studies have demonstrated a close association between lncRNAs and ubiquitination, with both entities playing a pivotal role in the modulation of signaling pathways involved in cancer regulation [159,160]. In the regulation of the drug resistance mechanism of gastric cancer, lncRNAs also participate in the regulation of protein ubiquitination. For instance, HULC was found to interact with FoxM1 and modulate the protein level of FoxM1 by inhibiting its ubiquitination process and stabilizing its expression, thereby mitigating cisplatin resistance in gastric cancer [67]. CRNDE in TAM-derived exosomes regulated PTEN expression by modulating NEDD4-1-mediated PTEN ubiquitylation, thus influencing the DDP resistance in GC [149]. OVAAL enhanced 5-FU resistance in GC by upregulating the levels of PC (pyruvate carboxylase) protein at the post-transcriptional level, thereby inhibiting the ubiquitin-mediated degradation of the PC protein [152]. In addition, the lncRNA CBSLR-YTHDF2 protein complex destabilized CBS mRNA by promoting the interaction between YTHDF2 and the m6A-modified CDS region of CBS mRNA, eventually leading to the high ubiquitination of the ACSL4 protein in GC cells [40] (Figure 3).

### 4.5. Drug Resistance-Related lncRNA-Mediated Regulation of Cell Signaling in GC

In addition to the above regulatory mechanisms of GC drug resistance, lncRNAs can also mediate drug resistance by managing the activity of downstream signaling pathways. It is well known that many signaling pathways have been found in cells, and they are very important for a cascade of biological reactions and gene expression. Extensive research suggests that drug resistance-related lncRNAs are closely related to various pathways directly or indirectly. In summary, these signaling pathways involved in GC resistance-related lncRNAs include the following: the wnt/β-catenin signaling pathway, the PI3K/AKT/GSK-3β signaling pathway, the NF-κB signaling pathway, the ERK1/2 signaling pathway, the Notch2 signaling pathway, the MAPK signaling pathway, and the p53 signaling pathway. The wnt/β-catenin signaling pathway is considered one of the most canonical signaling pathways, which manages the occurrence and development of tumors. Some studies have indicated that wnt co-current is over-activated in several kinds of solid tumors from humans, and the wnt/β-catenin signaling pathway may be a common pathway for many tumors. In DDP-resistant cell lines, BCAR4 [38], HOTAIR [62], and ZFAS1 [101] can activate the wnt/β-catenin signaling pathway by upregulating their expressions to increase the drug resistance of GC. Currently, the PI3K/AKT/mTOR signaling pathway is the focus of research. In addition to regulating cell growth and survival, protein transcription, glucose metabolism, and a variety of programmed cell death types such as apoptosis, autophagy, and iron death, the abnormal activation of this pathway affects the occurrence and progression of tumors including GC [161]. There are nine lncRNAs involved in the process of lncRNAs mediating the cellular mechanism of drug resistance in GC, namely, MALAT1 [79], CRAL [125], LOC101928316 [143], FOXD1-AS1 [52], ST7-AS1 [106], HOTAIR [62], SNHG12 [94], FAM84B-AS [48], and HIT000218960 [151]. These lncRNAs regulate the PI3K/AKT/mTOR signaling pathway through their abnormal expression. In addition to CRAL inhibiting the activity of the signaling pathway, the expression of the other six lncRNAs is upregulated to activate the signaling pathway, which ultimately lead to the occurrence of drug resistance in gastric cancer. The NF-κB signaling pathway refers to a variety of biological processes, including the development of cancer. Through a series of kinase activation and phosphorylation, NF-κB is activated [162,163]. As a transcription factor, NF-κB can bind to target genes in the nucleus and initiate the transcription of downstream genes. The NF-κB signaling pathway has been demonstrated to play a crucial role in regulating the expression of cytokines, immune-related receptors, and other factors, thereby influencing cell proliferation, apoptosis, and drug resistance [136,164]. In a cisplatin-resistant GC cell line, ASB16-AS1 can increase the activity of the NF-κB pathway by combining with ATM to promote TRIM37 phosphorylation, thus activating the resistance of GC [37]. As the first cell signaling pathway to be discovered, ERK1/2 is the core of many extracellular signals that promote cell proliferation [165]. Once ERK1/2 is abnormally activated, it will promote the proliferation and malignant transformation of cells and then cause abnormal expression of downstream target genes. In GC cells, a high expression level of BANCR was found to mediate gastric cancer cell cisplatin resistance by increasing the phosphorylation of the ERK protein [166]. In addition, DUSP5P1 promotes platinum resistance in gastric cancer cell lines by activating the ERK1/2 signaling pathway [45]. Regarding the role of drug resistance in gastric cancer, we also found that other signaling pathways are involved, such as the Notch signaling pathway and the p53 signaling pathway, by reviewing the documents selected for this review. In terms of the Notch signaling pathway, SNHG17 can positively regulate Notch2 protein expression through the ceRNA mechanism and then activate the Notch signaling pathway to induce drug resistance in gastric cancer [144]. Some lncRNAs that negatively regulate signaling pathways have also been found to regulate drug resistance mechanisms in gastric cancer. For example, TMEM44-AS1 can inhibit the p53 signaling pathway through the ceRNA mechanism to induce 5-FU drug resistance in gastric cancer [167].

Although a single lncRNA can mediate drug resistance through clear signaling pathways, the regulatory network between lncRNAs and signaling pathways plays a more important role. The clear regulatory axis also provides more evidence supporting the involvement of lncRNA regulatory signaling pathways in drug resistance mechanisms (Figure 4).

## 5. Metabolism, Tumor Microenvironment, and Drug Resistance-Related lncRNAs

Metabolism and the tumor microenvironment perform vital roles during carcinogenesis. Changes in the TME caused by the dysfunction of immune cells and vascular and stromal cells result in the hypoxia, high acid, and high interstitial fluid pressure environment around tumor cells [168,169]. Metabolism is closely related to the TME. Communication within the TME is dependent on tumor metabolic activity [170]. For instance, glycolytic metabolism of glucose (now known as the ”Warburg effect”) increases the level of lactic acid, thus acidifying the TME [171]. Tumor-associated hypoxia can further upregulate cellular glycolysis and lactic acid production, leading to an accumulation of acid and a change in the pH in the tumor microenvironment. Acidosis and hypoxia profoundly modulate cancer cell metabolism and disease progression to ensure significant metabolic reprogramming, which is beneficial to tumor progression [172]. Studies have indicated that lncRNAs function in metabolism and the TME, thus promoting carcinogenesis [173].

### 5.1. Metabolism and Drug Resistance-Related lncRNAs

Tumor cells are dependent on different metabolic networks including glucose, fatty acid, amino acid, and nucleic acid metabolism. It is widely known that lncRNAs play important roles in modulating metabolic processes [174]. In this review, several drug resistance-associated lncRNAs involved in metabolism were identified in GC. For lipid metabolism, fatty acid oxidation (FAO) is a vital process in the regulation of drug resistance. As a major pathway, FAO can promote fatty acid (FA) degradation and increase ATP and NADPH production [175]. In the process of FAO, FAs are important energy resources. The FAO process in mitochondria produces 2.5 times as much ATP compared with the glucose oxidation reaction [176]. Two lncRNAs are closely related to FAO metabolism and involved in the drug resistance of GC. The study by He et al. indicated that MACC1-AS1 is involved in the 5-Fu resistance of GC cells by regulating lipid metabolism [75]. In their study, He et al. found that the TGF-β1 secreted by MSCs could induce MACC1-AS1 expression in GC cells. After that, MACC1-AS1 could promote stemness and chemoresistance in FAO dependence [75]. Another study by Wu et al. pointed out that HCP5 contributed to stemness and drug resistance in GC by increasing the FAO reaction. Mechanistically, HCP5 and miR-3619-5p constitute the ceRNA regulatory network to upregulate PPARGC1A expression, which prompted FAO in GC cells [55]. For glucose metabolism in tumor cells, the Warburg effect plays an important role, even under aerobic conditions [177]. Some lncRNAs participate in the glucose metabolism in tumor cells through different mechanisms. Some lncRNAs affect the glycolysis process by regulating the expression of key enzymes of glycolysis and inducing drug resistance in gastric cancer cell lines; these include, for instance, SNHG7 [137], HAGLR [54], and SNHG1 [107]. SNHG7 desensitizes gastric cancer cells to cisplatin via the miR-34a/LDHA-glycolysis axis [137]. SNHG16 decreases miR-506-3p by sponging it, forming a ceRNA network to regulate PTBP1 expression in GC cells, thus mediating expressions of multiple glycolysis enzymes and including GLUT1, HK2, and LDHA [138]. HAGLR can function in oncogenic roles by sponging miR-338-3p to activate the LDHA-glycolysis pathway in the 5-Fu resistance of GC cells [54]. In Taxol-resistant cell line, SNHG1 can act as a ceRNA of miR-216b-5p to upregulate HK2, a glucose metabolism key enzyme, thus targeting the HK2–glycolysis axis and promoting Taxol resistance [107]. In addition to the lipid metabolism and glucose metabolism involved in drug resistance mechanisms, lncRNAs also affect nucleotide metabolism in gastric cancer. For instance, Tan et al. indicated that GC cells with high expression of OVAAL were more resistant to 5-FU and became resistant to 5-FU treatment. OVAAL binds to PC and stabilizes PC against HSC70/CHIP-mediated cytoplasmic ubiquitination and degradation in the cytoplasm. The above process eventually triggers the production of oxaloacetate from pyruvate and the following accumulation of malate and aspartate, leading to 5-Fu resistance [152] (Figure 5).

### 5.2. Tumor Immune Microenvironment and Drug Resistance-Related lncRNAs

The tumor immune microenvironment covered in the TME has been proven as a special microenvironment that can recombine the cancer biology process. The changes in the tumor immune microenvironment finally affect cancer prognosis and response to drug treatment [178]. The tumor immune microenvironment contains not only tumor cells but also myeloid cells and lymphocytes. These infiltrations of immune cells can exchange immune phenotypes in the TME and influence immune escape and therapy resistance [179,180]. Studies have shown that lncRNAs not only affect protein expression involved in the immune response through different mechanisms but can also regulate the functional maintenance of immune cells, thus facilitating the escape of tumor cells from immune surveillance [181]. In this review, several drug resistance-associated lncRNA were found to be involved in immune regulation in GC. (1) Regulation of the programmed cell death protein 1 (PD)-L1 protein expression and induction of drug resistance: As a ligand of PD-1, PD-L1 is mainly synthesized from activated T and B cells and acts as a co-suppressor receptor molecule, which can inhibit the function of PD-1-expressing T cells and suppress the immune response [182,183]. PD-L1 is often observed to have a high-level status and is associated with immune evasion of cancer cells [184]. Two lncRNAs regulate PD-L1 expression by means of different mechanisms to affect the immune response and then induce the drug resistance of GC. Dang et al. indicated that SNHG15 could absorb the miR-141 increasing PD-L1 expression, which promoted the resistance of GC cells to immune therapy [139]. In gastric cancer, SNHG15 is usually highly expressed. Through the regulatory mechanism of ceRNA, upregulated SNHG15 positively promotes the expression of PD-L1 on the surface of dendritic cells, macrophages, or gastric cancer cells in gastric cancer tissues, induces apoptosis, non-response, and dysfunction of T cells, and eventually leads to an increase in the immune escape activity of gastric cancer cells and promotes immunotherapy resistance [139]. NUTM2A-AS1 positively regulates the expression of target genes *TET1* and *HIF-1A* by forming ceRNA with miR-376a. TET1 can combine with HIF-1A to modulate PD-L1 expression positively. Once the expression level of PD-L1 is upregulated, the immune escape function of gastric cancer cells is increased, which promotes the drug resistance of immunotherapy in gastric cancer [85]. (2) M2 macrophage polarization: As the major component of the TME, TAMs play an essential role in the occurrence, development, metastasis, and chemoresistance of tumors [185]. In patients with GC, TAMs are polarized to the M2 phenotype and contribute to tumor cell proliferation, resistance, and a poor prognosis [186]. In GC cells, HIF1A-AS2 and RP11-366L20.2 both absorb miR-29c, resulting in the upregulation of the *LOX* gene [134]. Studies have shown that *LOX* family members are involved in the establishment and maturation of the tumor microenvironment [187]. A high level of LOX can facilitate macrophage polarization toward the M2 phenotype, which results in the immune escape of cancer cells and drug resistance in GC [134]. In addition, M2-polarized macrophages can secrete CRNDE-rich exosomes to facilitate NEDD4-1-mediated PTEN ubiquitination, resulting in drug resistance in GC [149] (Figure 5).

### 5.3. “Cross-Talk” in TME and Drug Resistance-Related lncRNAs

PhD Whiteside pointed out that there is a close cross-talk among various cells including tumor cells in the TME. This important medium of close communication in the TME mainly relies on small vesicles secreted by cells (also called exosomes). As a major component of the TME, MSCs play a vital role in facilitating tumor progression. These small vesicles induced by tumor cells can re-program the functional profile of MSCs from normally trophic to pro-tumorigenic. At the same time, MSCs affected by cancer cell signals also give back to tumor cells by producing their own exosomes carrying and delivering molecular signals, further promoting the malignant phenotype of tumor cells [188]. In this review, we also found that several lncRNAs play important roles in this cross-talk and induce drug resistance in GC. For instance, one study found that MSCs secreted TGF-β1, which induced MACC1-AS1 expression and promoted FAO-dependent stemness and chemoresistance [75]. In MDR GC cell lines, MSCs were found to secret exosomes containing HCP5 to confer chemo-resistance and enhance the FAO-dependent stemness of GC cells by adsorbing miR-3619-5p to increase PPARGC1A expression, finally leading to the transactivation of CPT1 by the PGC1α/CEBPB complex [55]. In DDP-resistant cells, GC cells secreted high levels of exosomal FGD5-AS1 and transmitted these exosomes to parental cells, thus inducing DDP resistance by sponging miR-195 [51]. In addition, M2 macrophages were also found to secret exosomes containing CRNDE to facilitate NEDD4-1-mediated PTEN ubiquitination, thus inducing the CDDP resistance of GC cells [149] (Figure 5).

## 6. Clinical Application of Drug Resistance-Related lncRNAs

Drug resistance-associated lncRNAs take part in regulating drug resistance through a variety of mechanisms in GC. At the same time, these lncRNAs also play an important role in clinical diagnosis, therapeutic evaluation, and the prediction of therapeutic efficacy. (1) Diagnosis: In paclitaxel-resistant cell lines, the expression of PVT1 can indicate whether there is paclitaxel resistance and lymph node metastasis [189]. Chen et al. applied the non-negative matrix factorization (NMF) algorithm using the TCGA database to identify a new cluster of survival-related GC and found that ZFPMA-AS1 could regulate TIME and drug sensitivity associated with anticancer treatment. Patients with high ZFPM2-AS1 expression had worse survival than those with low ZFPM2-AS1 expression in STAD [190]. (2) Prognostic risk assessment and efficacy prediction: Compared with the limitations of a single marker in the diagnosis and prediction of GC, a gene set or model prediction is currently widely used in clinical practice. In this review, a total of 17 articles (Table 3) used the differential expression of different functionally related lncRNAs to construct algorithms, so as to provide a basis for guiding clinical prognosis and predicting therapeutic effects. Four articles [191,192,193,194] used drug-resistant tissues or cells (PR, MDR, and DCSR) for differential lncRNA screening to construct a model. Four articles [193,194,195,196] constructed prediction models based on pyroptosis-related lncRNAs (PRlncRNAs). These research studies all conducted the Least Absolute Shrinkage and Selection Operator (LASSO) algorithm to construct the PRlncRNA model. The data in a high-risk group indicated that these patients had adverse prognoses compared with a low-risk group. The high-risk group patients always prompted a lower tumor mutation burden and gene mutation frequency, which predicted prognosis and immunotherapy or chemotherapy drug sensitivity. Two articles [197,198] constructed prediction models based on stemness-related lncRNAs (SRlncRNAs). In the models, a higher score might hint at a better performance in predicting therapy response including immunotherapy and chemotherapy. Three articles [129,199,200] constructed prediction models based on ferroptosis- or cuproptosis-related lncRNAs (F/CRlncRNAs). These models were usually established to be used to predict the prognosis of GC. Patients with a high-risk score were associated with immune escape based on integrated bioinformatics analyses including low genomic instability, low tumor mutation burden (TMB), and worse immunotherapy response. The evaluation of genomic instability, which is closely related to the efficacy of immunotherapy, was also studied in two articles [201,202]. Using RNA sequencing and single nucleotide variant (SNV) data from TCGA datasets, the authors established the genomic instability-associated lncRNA signature (GILncSig) based on the accumulation of gene mutation counts to evaluate chemotherapy drug sensitivity and immune landscape changes, providing a basis for clinical immunotherapy efficacy evaluation. In addition, some algorithm-based models using other functionally related lncRNAs, such as autophagy-related lncRNAs (ARlncRNAs) and PLT-related lncRNAs (PLTRlncRNAs), have been applied to clinical prediction. For example, a nine-optimal gene risk model based on ARlncRNAs was constructed, and the results indicated that high-risk patients gained higher PD-1/PD-L1 expressions and higher sensitivity to chemotherapy agents compared with the low-risk group [203]. Another prediction model based on PLTRlncRNAs data suggested that high-risk individuals had a poorer prognosis because of a low infiltration of immune cells and a poor response to immunotherapy [204] (Table 3). Unfortunately, these models are limited to theoretical predictions and are not applied in clinical practice. This also provides more ideas for subsequent researchers to conduct in-depth research in related fields.

In addition to their diagnostic and prediction functionalities, the aberrant expression of certain drug resistance-associated lncRNAs identified in this study exhibits a significant correlation with clinicopathological characteristics and prognostic outcomes among patients diagnosed with gastric cancer. In terms of clinicopathological characteristics, the high expression levels of six drug resistance-associated lncRNAs, namely, ADAMTS9-AS2 [32], BCAR4 [38], FGD5-AS1 [51], PANDAR [86], SLCO1C1 [91], and UCA1 [97], were found to be positively correlated with tumor size. The expression of four lncRNAs including BCAR4 [38], FAM84B-AS [48], SLCO1C1 [91], and UCA1 [97] is closely associated with histological grade and differentiation. The expression of some drug resistance-associated lncRNAs is also closely related to Lauren classification and Borrman type. These lncRNAs include BCAR4 [38] and UCA1 [97]. Just as we introduced the close relationship between drug-resistant lncRNA and the malignant phenotype of gastric cancer at the cellular level, in terms of the clinical tissue expression level, the abnormal expression of some drug-resistant lncRNAs is also closely related to neurovascular invasion, lymphatic metastasis, and distant metastasis. These lncRNAs include ADAMTS9-AS2 [32], BCAR4 [38], FAM84B-AS [48], and UCA1 [97,99]. As shown in Table 4, there are 15 lncRNAs whose abnormal expression is closely related to the TNM stage.

In terms of prognosis evaluation, the expression of some drug-resistant lncRNAs in this review is also closely related to poor prognosis. The aberrant expression of 22 lncRNAs (ABL, ADAMTS9-AS2, ARHGAP5-AS, CBSLR, CRART16, CRNDE, DUSP5P1, D63785, FAM84B-AS, FEZF1-AS1, HCP5, HULC, LINC-PINT, MACC1-AS1, MALAT1, NUTM2A-AS1, PANDAR, PVT1, PITPNA-AS1, SLCO1C1, SNHG12, SUMO1P3, and UCA1) were found to be significantly associated with overall survival (OS). The expression levels of three lncRNAs (ARHGAP5-AS, DUSP5P1, and PVT1) were found to be correlated with progression-free survival (PFS), while another four lncRNAs (CBSLR, CRNDE, MACC1-AS1, MALAT1) showed a significant association with disease-free survival (DFS). Additionally, the expression patterns of three lncRNAs were linked to recurrence-free survival (RFS) (BCAR4 and EIF3J-DT) and tumor recurrence (LINC-PINT). Furthermore, nine lncRNAs (ABL, ARHGAP5-AS, BCAR4, DUSP5P1, FAM84B-AS, MACC1-AS1, MALAT1, PANDAR, and UCA1) exhibited potential as independent prognostic factors for predicting adverse outcomes in gastric cancer. The above results are detailed in Table 4.

In this review, most of the identified lncRNAs that regulate the drug resistance of gastric cancer may become therapeutic targets for clinical application in the future. The majority of these lncRNAs, which exert a crucial role in the regulation of drug resistance in gastric cancer, mediate their effects on target genes through mechanisms such as vector-mediated overexpression and RNAi (RNA interference) methods (siRNA and shRNA). Ultimately, this leads to enhanced chemosensitivity, suppression of drug resistance, and inhibition of malignant phenotypes. LncRNAs, which play the role of tumor suppressor genes in the regulation of drug resistance in gastric cancer, can increase the sensitivity of chemotherapy drugs and improve the therapeutic effect under the action of overexpression vectors. RNAi has become an important means for studying gene function in mammalian cells. The downregulation effect of target genes is achieved through the action of exogenous double-stranded RNA or short hairpin RNA (shRNA) molecules [205,206]. The CRISPR-Cas9 system has been discovered as a groundbreaking genome editing tool for use in cancer treatment to suppress and activate long non-coding RNA [207]. As shown in Appendix A, the knockdown of PANDAR by the CRISPR-Cas9 system inhibited multidrug resistance in GC [86]. Furthermore, RNAi technology can effectively inhibit certain lncRNAs associated with drug resistance in gastric cancer such as HOTTIP [65] and FGD5-AS1 [51], which play a crucial role in the delivery of exosomes. The lncRNAs that play pivotal roles in drug resistance hold promising potential as therapeutic targets for overcoming GC drug resistance (Appendix A).

## 7. Conclusions and Perspectives

Although the current treatment drugs and regimens for advanced GC are continuously updated, there has been limited improvement in overall patient outcomes over the past few decades. Chemoresistance remains a major obstacle to achieving effective treatment in these patients. Whether intrinsic or acquired, drug resistance is a complex and multifactorial process closely associated with cancer cells and the tumor microenvironment [208]. The molecular mechanism of chemoresistance in GC has garnered increasing attention from researchers. Currently, it is established that lncRNAs play crucial regulatory roles in various physiological and pathological processes, including chemoresistance. Accumulating evidence suggests that lncRNAs are implicated in the regulation of drug resistance in gastric cancer through diverse mechanisms. In this comprehensive and systematic review, we aimed to elucidate the role of lncRNAs in drug resistance comprehensively. These lncRNAs associated with drug resistance exert significant effects on the proliferation, invasion, metastasis, and apoptosis of drug-resistant GC cell lines because of their aberrant expression patterns. With drug resistance-related lncRNAs as a central focus, multiple mechanisms encompassing upstream and downstream regulations mediate drug resistance in GC. Within this intricate regulatory network, alterations in numerous molecules and microenvironments contribute to some extent toward the development of GC drug resistance. Furthermore, different prediction models based on high-throughput sequencing results also provide a clinical foundation for guiding future drug therapy for GC.

Although the involvement of lncRNAs in GC drug resistance has been extensively investigated for several decades, significant progress has been made only in understanding a fraction of the underlying mechanisms. At the same time, in order to improve the overall survival of patients with gastric cancer as much as possible with the application of new drugs or off-label drugs in the treatment of gastric cancer, more mechanisms need to be discovered. For instance, lncRNAs are also involved in the regulation of bevacizumab in GC drug resistance [41]. Highly expressed CRART16 in gastric cancer tissues can adsorb miR-122-5p and upregulate the expression of downstream oncogene FOS through the ceRNA mechanism. A high level of FOS expression, which eventually leads to the upregulation of VEGFD expression, inhibits the inhibitory effect of bevacizumab on gastric cancer cells. In addition, in the intricate TME, elucidating the precise role of lncRNAs in regulating GC drug resistance is inevitably confronted with numerous challenges. However, these research gaps also offer abundant opportunities for future investigations. For instance, recent discoveries highlighting metal-dependent programmed cell death (PCD)-related lncRNAs hold great promise for advancing personalized care based on chemosensitivity models of GC cells and facilitating the development of novel therapeutic strategies to overcome chemoresistance. Overall, it is crucial to recognize that drug resistance in GC represents a dynamic process closely intertwined with changes occurring within the TME. Unraveling the intricate involvement of lncRNAs in this process remains an imperative direction for future researchers.

Below is a simple summary of this review and some remaining open questions:Gastric cancer, being one of the most prevalent malignant neoplasms, significantly impacts global population health.Drug resistance poses a significant impediment to achieving optimal therapeutic outcomes in the treatment of malignant tumors.Long non-coding RNAs (lncRNAs) play a pivotal role in a wide range of pathological and physiological processes, encompassing the regulation of drug resistance.What special role do drug-resistant lncRNAs play in the malignant phenotype of gastric cancer?Can the ceRNA model be considered as the exclusive regulatory mechanism utilized by lncRNAs in conferring drug resistance to gastric cancer?What is the functional significance of lncRNAs among the tumor microenvironment, metabolism, and drug resistance mechanisms in gastric cancer?How are drug-resistant lncRNAs used to guide the diagnosis and treatment of gastric cancer?

## Figures and Tables

**Figure 1 biomolecules-14-00608-f001:**
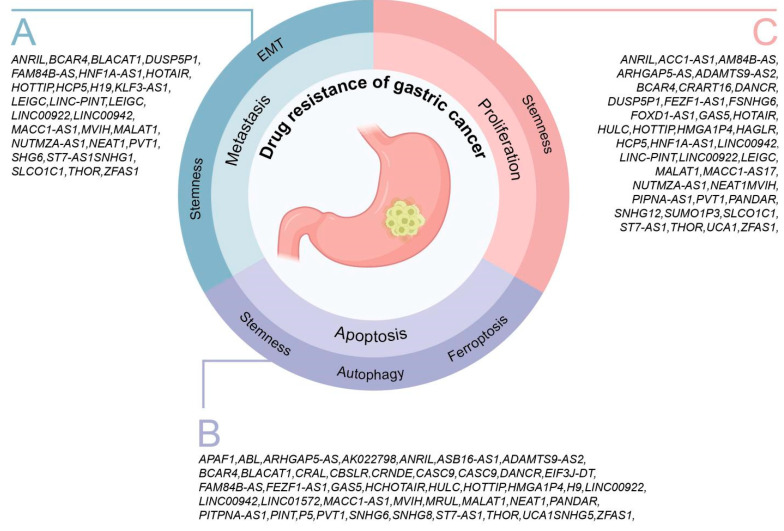
Overview and malignant phenotypes of drug resistance-related lncRNAs in GC. Zone A: Metastasis and drug resistance-related lncRNAs. Zone B: Apoptosis and drug resistance-related lncRNAs. Zone C: Proliferation and drug resistance-related lncRNAs.GC: gastric cancer.

**Figure 2 biomolecules-14-00608-f002:**
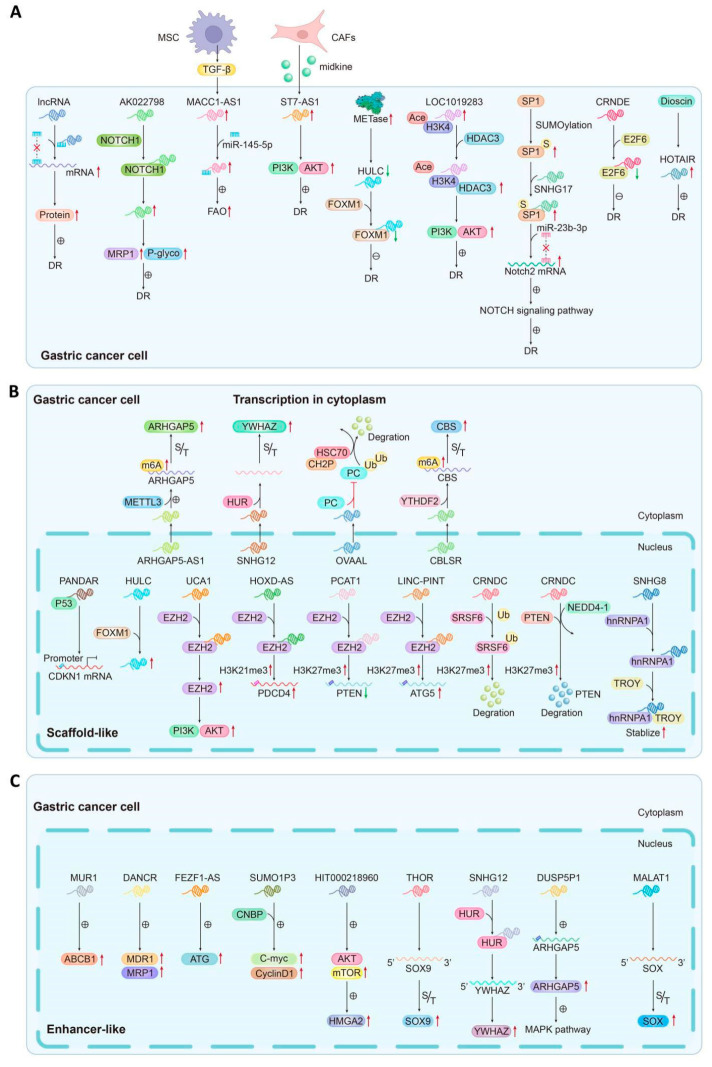
Upstream and downstream regulation mechanisms of drug resistance-related lncRNAs. (**A**) Upstream regulation of drug resistance-related lncRNAs (DR: drug resistance; FAO: fatty acid oxidation). (**B**) Downstream regulation of drug resistance-related lncRNAs on scaffold-like lncRNAs in the nucleus and transcription/translation functions of lncRNAs in the cytoplasm. (**C**) Downstream regulation of drug resistance-related lncRNAs on enhancer-like lncRNAs in the nucleus (S/T: stabilize/transcribe).

**Figure 3 biomolecules-14-00608-f003:**
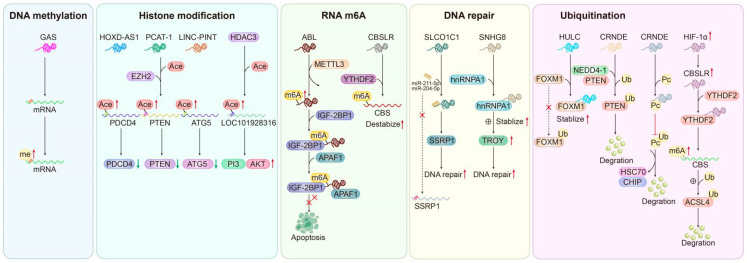
Epigenetics and drug resistance-related lncRNAs including DNA methylation, histone modification, RNA m6A modification, DNA repair, and ubiquitination.

**Figure 4 biomolecules-14-00608-f004:**
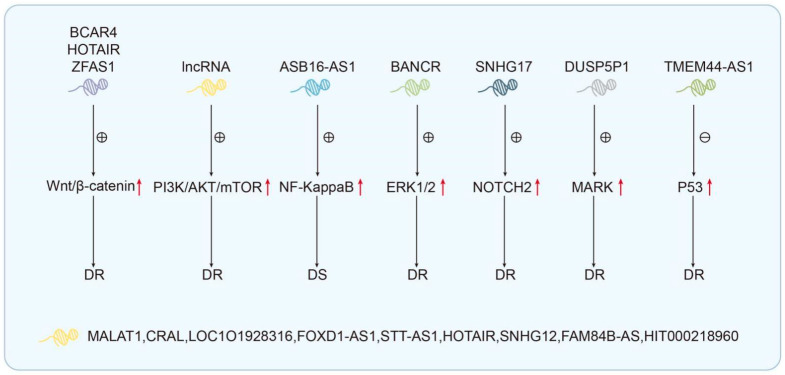
Drug resistance-related lncRNA-mediated regulation of cell signaling (DR: drug resistance; DS: drug sensitivity).

**Figure 5 biomolecules-14-00608-f005:**
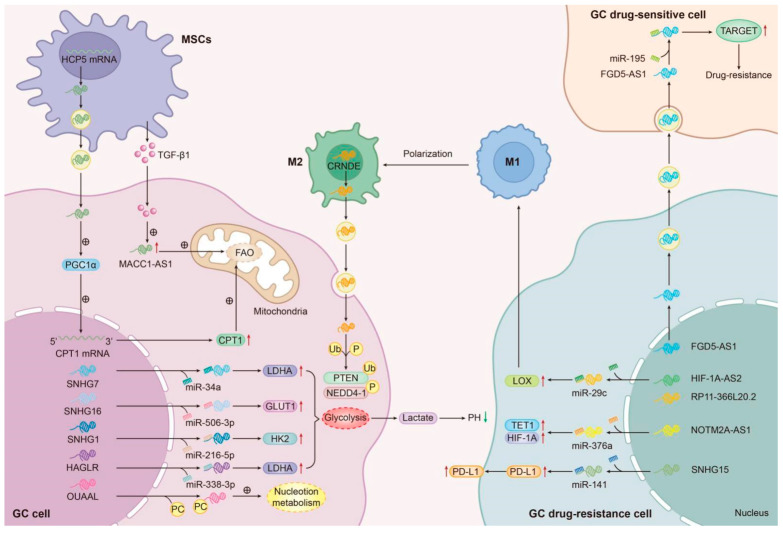
The ”cross-talk” among metabolism, tumor microenvironment, and drug resistance-related lncRNAs.

**Table 1 biomolecules-14-00608-t001:** Drug-resistant lncRNAs and their functions in gastric cancer.

LncRNA	Resistance (Sensitivity)	Functions	Refs.
ABL	paclitaxel, 5-FU	apoptosis	[31]
ADAMTS9-AS2	(cisplatin)	proliferation, apoptosis	[32]
AK022798	cisplatin	apoptosis	[33]
ANRIL	cisplatin, 5-FU	proliferation, metastasis, apoptosis	[34,35]
ARHGAP5-AS	cisplatin, 5-FU, rapamycin,	proliferation, apoptosis	[36]
	doxorubicin, actinomycin D		
ASB16-AS1	cisplatin	proliferation, stemness	[37]
BCAR4	cisplatin	stemness	[38]
BLACAT1	oxaliplatin	metastasis, apoptosis	[39]
CBSLR	cisplatin	ferroptosis	[40]
CRART16	bevacizumab	proliferation	[41]
CRNDE	oxaliplatin, 5-FU	apoptosis, autophagy	[42,43]
DANCR	cisplatin	proliferation, apoptosis	[44]
DUSP5P1	oxaliplatin	proliferation, metastasis	[45]
D63785	doxorubicin	proliferation, metastasis	[46]
EIF3J-DT	oxaliplatin, 5-FU	apoptosis, autophagy	[47]
FAM84B-AS	cisplatin	proliferation, metastasis, apoptosis	[48]
FEZF1-AS1	cisplatin, 5-FU	proliferation, autophagy	[49]
FGD5-AS1	5-FU	proliferation	[50,51]
FOXD1-AS1	cisplatin	proliferation	[52]
GAS5	(adriamycin)	proliferation, apoptosis	[53]
HAGLR	5-FU	proliferation	[54]
HCP5	oxaliplatin, cisplatin,5-FU	proliferation, apoptosis, stemness	[55,56,57]
HMGA1P4	cisplatin	proliferation, apoptosis	[58]
HNF1A-AS1	5-FU	proliferation, metastasis	[59]
HOTAIR	cisplatin, oxaliplatin	proliferation, metastasis, apoptosis	[60,61,62,63,64]
HOTTIP	cisplatin	proliferation, metastasis, apoptosis	[65]
HULC	cisplatin	proliferation, apoptosis, autophagy	[66,67]
H19	adriamycin	apoptosis	[68]
KLF3-AS1	cisplatin	proliferation, metastasis	[69]
LEIGC	5-FU	proliferation, metastasis	[70]
LINC01572	cisplatin	autophagy	[71]
LINC-PINT	cisplatin	proliferation, metastasis, autophagy	[72]
LINC00922	cisplatin	proliferation, metastasis, apoptosis	[73]
LINC00942	cisplatin	apoptosis, stemness	[74]
MACC1-AS1	5-FU	stemness	[75]
MALAT1	cisplatin, oxaliplatin, vincristine	proliferation, metastasis, apoptosis, autophagy, stemness	[76,77,78,79,80,81]
MRUL	doxorubicin	proliferation, apoptosis	[82]
MVIH	gemcitabine	proliferation, metastasis, apoptosis	[83]
NEAT1	adriamycin	proliferation, metastasis, apoptosis	[84]
NUTM2A-AS1	PD-1	proliferation, metastasis	[85]
PANDAR	oxaliplatin, 5-FU	proliferation, apoptosis	[86]
PITPNA-AS1	cisplatin	proliferation, apoptosis	[87]
PVT1	cisplatin, 5-FU	proliferation, metastasis, apoptosis	[88,89,90]
SLCO1C1	oxaliplatin	proliferation, metastasis	[91]
SNHG1	paclitaxel	metastasis	[54]
SNHG5	cisplatin	apoptosis	[92]
SNHG6	cisplatin	proliferation, metastasis, apoptosis	[93]
SNHG12	oxaliplatin, 5-FU	proliferation	[94]
ST7-AS1	cisplatin	proliferation, metastasis, apoptosis	[93]
SUMO1P3	cisplatin, 5-FU	proliferation, metastasis	[95]
THOR	cisplatin	stemness	[96]
UCA1	cisplatin, doxorubicin	proliferation, apoptosis	[97,98,99,100]
ZFAS1	cisplatin, paclitaxel	proliferation, metastasis, apoptosis	[101]

**Table 3 biomolecules-14-00608-t003:** Clinical application of drug resistance-related lncRNAs in gastric cancer.

LncRNA or Model	LncRNA Name or ID	Functions	Data	Experimental Methods	Clinical Application	Refs.
A 9-ARlncRNA signature	AL357054.4, AC018682.1, A2M-AS1, AP001107.5, CAPN10-DT, HAND2-AS1, LINC01081, PIK3CD-AS1, ZNF710-AS1	ARlncRNAs	TCGA	HTS, algorithm	predicting prognosis and efficacy of immunotherapy and chemotherapy	[203]
A 13-SRlncRNA signature	AC026369.2, AC024267.4, AC017074.1, AC0104695.4, AC016394.3, AC009022.1, AC112484.3, AC005391.1, LINC00941, LINC02532, LINC01614, LINC01943, SMIM25	SRlncRNAs	TCGA	HTS, algorithm	predicting immunotherapy response	[197]
A 23-SRlncRNA signature	AP000873.4, AC116158.1, RNF144A-AS1, LINC01094, MAPKAPK5-AS1, AL136115.1, AL391152.1, AC147067.2, AL356215.1, ADAMTS9-AS1, AC011747.1, AL353796.1, AC104695.4, AC087521.1, AC078860.2, AC027682.6, AC104809.2, AC129507.1, AC010768.2, AC026412.3, LINC01614, LINC00519, LINC00449	SRlncRNAs	TCGA/Zhongshan/IMvigor210	HTS, algorithm	predicting chemotherapy and immunotherapy response	[198]
A 6-GIRlncRNA signature	AC010789.1, HOXA10-AS, LINC02678, LINC01150, RHOXF1-AS1, TGFB2-AS1	GIRlncRNAs	TCGA	HTS, algorithm	predicting immunotherapy response	[201]
A 17-FRlncRNA signature	AC104260.2, AP000438.1, AL022316.1, AL391152.1, AC021106.3, AC131391.1, AL355001.1, AP000695.1, AP001107.6, AC007391.1, AL021154.1, AC104758.1, FP700111.1, MACORIS, RFS1-IT2, SPATA13-AS1, SCAT8	FRlncRNAs	TCGA/GEO	HTS, algorithm	predicting prognosis and therapeutic response	[129]
A 14-PRlncRNA signature	AC074286.1, AC013275.2, C10orf91, CTD-2377D24.6, LINC00607, LINC01094, LINC00607, LINC01588, MMP25-AS1, MLLT4-AS1, RP3-522D1.1, RP11-61A14.1, TUSC8, TRPM2-AS	PRlncRNAs	TCGA/GEO	HTS, algorithm	predicting differential sensitivity to multiple chemotherapeutic agents	[195]
A CElncRNA-GC1 and AJCC stage	lncRNA-GC1	MRRlncRNAs	Central data	HTS, algorithm	predicting prognosis and chemotherapy response after surgery	[191]
A 12-F/CRlncRNA signature	ENSG00000221819.5, ENSG00000230387.2, ENSG00000233262.1, ENSG00000239265.4, ENSG00000241111.1, ENSG00000248279.4, ENSG00000248356.1, ENSG00000249807.1, ENSG00000250303.3, ENSG00000256220.1, ENSG00000265194.1, ENSG00000266957.1	F/CRlncRNAs	TCGA and literature	HTS, algorithm	predicting chemotherapy response	[199]
A 3-PRlncRNA signature	AC017076.1, CYMP-AS1, PVT1	PRlncRNAs	GSEA and literature	HTS, algorithm	predicting immunotherapy and chemotherapy drug sensitivity	[193]
A 10-CRlncRNA signature	AC016737.1, AL391152.1, AL121748.1, AL512506.1, AC104809.2, AL353804.2, AL353796.1, AL355574.1, LINC01980, TYMSOS	CRlncRNAs	TCGA	HTS, algorithm	predicting prognosis and presenting immune landscape	[200]
A 4-PRlncRNA signature	HAND2-AS1, LINC01354, PGM5-AS1, RP11-276H19.1	PRlncRNAs	TCGA	HTS, algorithm	predicting prognosis and immune microenvironment status	[194]
A 11-PRlncRNA signature	AL353804.1, AC147067.2, AP001318.2, AC018752.1, ACTA2-AS1, AL121772.1, AC005332.4, AC245041.2, HAGLR, RRN3P2, UBL7-AS1	PRlncRNAs	TCGA	HTS, algorithm	predicting prognosis and immune landscape	[196]
A 8-GIRlncRNA signature	AC078883.2, AL049838.1, AL359182.1, AL365181.3, LINC01436, LINC01833, LINC01614, RHOXF1-AS1	GIRlncRNAs	TCGA/GEO	HTS, algorithm	predicting prognosis and immunotherapy response	[202]
A 7-PLTRlncRNA signature	AC002401.4, AC129507.1, AL513123.1, AL355574.1, AL356417.2, LINC01697, LINC01094	PLTRlncRNAs	TCGA	HTS, algorithm	predicting prognosis and immunotherapy response	[204]
A 11-DCSRlncRNA signature	AC007277.1, AC005324.4, AL512506.1, AC068790.7, AC022509.2, AC113139.1, LINC02532, LINC00106, AC005165.1, MIR100HG, UBE2R2-AS1	DCSRlncRNAs	TCGA/GEO	HTS, algorithm	predicting chemotherapy response and immune infiltration in patients with GC	[192]
PVT1	PVT1	PRRlncRNAs	DE SGC7901/SGC7901P	HTS	predicting lymph node invasion	[189]
ZFPM2-AS1	ZFPM2-AS1	IRlncRNAs	TCGA	HTS, algorithm	predicting survival and reducing the sensitivity to cisplatin	[190]

Abbreviations: DElncRNA: differentially expressed lncRNA; ARlncRNA: autophagy-related lncRNA; SRlncRNA: stemness-related lncRNA; GIRlncRNA: Genomic instability-related lncRNA; FRlncRNA: ferroptosis-related lncRNA; PRlncRNA: pyroptosis-related lncRNA; CRlncRNA: cuproptosis-related lncRNA; PLTRlncRNA: platelet-related lncRNA; TCGA: The Cancer Genome Atlas; GEO: Gene Expression Omnibus; HTS: high-throughput sequencing; DCSRlncRNA: docetaxel, cisplatin, and S-1-related lncRNA.

**Table 4 biomolecules-14-00608-t004:** Relationship between drug resistance-associated lncRNA and prognosis of gastric cancer.

LncRNA	Sources	Expression	Prognosis	Refs.
ABL	Microarray (T vs. no-T)	↑	TNM(+), OS(+), IPF	[31]
ADAMTS9-AS2	Literature search	↑	Size(+), lymphatic invasion(+), TNM(+), OS(+)	[32]
ARHGAP5-AS	Microarray (R-c vs. S-c)	↑	Gender(+), TNM(+), OS(+), PFS(+), IPF	[36]
BCAR4	Literature search	↑	Size(+), Lauren type(+), histological grade(+), lymph node metastasis(+), distant metastasis(+), TNM(+), 3-y RFS(+), IPF	[38]
CBSLR	Microarray (H vs. no-H)	↑	OS(+), DFS(+)	[40]
CRART16	Microarray (T vs. no-T)	↑	TNM(+), OS(+)	[41]
CRNDE	Literature search	↓	OS(+), DFS(+)	[42]
DUSP5P1	CHIP-sequencing	↑	TNM(+), OS(+), PFS(+), IPF	[45]
D63785	Microarray (T vs. no-T)	↑	3-y OS(+)	[46]
EIF3J-DT	Microarray (R-c vs. S-c)	↑	RFS(+)	[47]
FAM84B-AS	Microarray (R-t vs. S-t)	↑	Differentiation(+), vascular cancer thrombus(+), nerve invasion(+), TNM(+), IPF	[48]
FEZF1-AS1	Literature search	↑	OS(+)	[49]
FGD5-AS1			Size(+), TNM(+)	[51]
HCP5	Literature search	↑	5-y OS(+)	[56]
HULC	Literature search	↑	5-y OS(+)	[66]
LINC-PINT	Literature search	↓	Recurrence(+), OS(+)	[72]
MACC1-AS1	Literature search	↑	TNM(+), DFS(+), OS(+), IPF	[75]
MALAT1	Literature search	↑	OS(+), DFS(+)	[76]
	TCGA (T vs. no-T)	↑	OS(+), IPF	[79]
NUTM2A-AS1	Literature search	↑	TMN(+), OS(+)	[85]
PANDAR	Microarray (T vs. no-T)	↑	Size(+), TNM(+), 5-y OS(+), IPF	[86]
PVT1	TCGA/GEO (T vs. no-T)	↑	TNM(+), OS(+), PFS(+)	[89]
PITPNA-AS1	Microarray (T vs. no-T)	↑	OS(+)	[87]
SLCO1C1	Microarray (T vs. no-T)	↑	Size(+), differentiation(+), OS(+)	[91]
SNHG12	Literature search	↑	TNM(+), OS(+)	[94]
SUMO1P3	Literature search	↑	TNM(+), OS(+)	[95]
UCA1	Microarray (T vs. no-T)	↑	Size(+), differentiation(+), Borrman type(+), Lauren type(+), invasion(+), TNM(+), IPF	[97]
	TCGA/GEO (T vs. no-T)	↑	Lymph node metastasis(+), distant metastasis(+), TNM(+), 5-y OS(+), IPF	[99]
	GEO (T vs. no-T)	↑	3-y OS(+)	[100]

Abbreviations: T: tumor; no-T: no-tumor; R-t: drug resistance tissue; S-t: drug sensitivity tissue; R-c: drug resistance cell line; S-c: drug sensitivity cell line; TNM: TNM stage; OS: overall survival; PFS: progress-free survival; DFS: disease-free survival; RFS: recurrence-free survival; IPF: independent prognostic factor; (+): positive correlation; ↑: high expression; ↓: low expression.

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
