# Peer review of "Long Non-Coding RNAs in Drug Resistance of Gastric Cancer: Complex Mechanisms and Potential Clinical Applications"

_biomolecules, 2024, doi:10.3390/biom14060608_

Round 1

Reviewer 1 Report

Comments and Suggestions for Authors

In this comprehensive review, the author provides a thorough summary of the current knowledge surrounding long non-coding RNAs (lncRNAs) in drug resistance of gastric cancer (GC). This review aims to summarize the molecular mechanisms underlying GC drug resistance regulated by lncRNAs and also discuss the potential clinical applications of lncRNAs as biomarkers and therapeutic targets in GC. This article highlights the importance of understanding the role of lncRNAs in GC, particularly in the context of the Special Role of Drug-Resistant lncRNAs, CeRNA Model and Regulatory Mechanisms, Functional Significance in Tumor Microenvironment, Metabolism, and Drug Resistance, Utilization for Diagnosis and Treatment Guidance. However, a noteworthy reference is the insightful review article in Frontiers in Oncology (2022, 12: 841411), which focused on similar topics. To enhance the value of this review and contribute novel insights, the author should incorporate the following comments:

1.     Consider using bullet points to highlight key information in the simple summary for improved readability and comprehension.

2.     Line 59: Replace "antioncogenes" with "tumor suppressor."

3.     Consider whether the cartoon illustrating the gene motif is necessary for the review's content and purpose. If not, it may be omitted.

4.     Change the data series based on the resistance column. Replace "multi-drug" with specific drugs. Include HOTAIR and CAS9 involvement in Adriamycin resistance (Table 1). Provide more description about PD-1 and bevacizumab resistance, as these have not been discussed in the referenced article (Frontiers in Oncology).

5.     Modify the results series based on the Functions column for clarity and consistency (Table 2).

6.     Few abbreviations were missing in the text. Please provide abbreviations for DR (Drug Resistance) and FAQ in the figure legend for reader understanding (Fig. 2).

7.     Include each lncRNA mentioned, such as A9-ARlncRNA, in the table for completeness. Replace "ARlncRNAs" with "autophagy-resistant" and "SRlncRNAs" with "stemness-resistant" for clarity (Table 2).

8.     Correct all mistakes in the text, including ensuring consistent spacing between sentences.

Comments on the Quality of English Language

Minor editing of English language required

Author Response

March 25, 2024

Editor-in-Chief,

Biomolecules,  

Dear Editor,  

    Thank you very much for your letter and the reviewers’ very constructive comments. We recognized that you and the reviewers kindly spent a lot of time and provided expert comments.  

    In response to the comments, we have made point-by-point modifications to address all the comments, and the details of modifications are described in response letter to reviewers.

    This manuscript has not been published, and is not under consideration elsewhere.  

    All the authors have contributed significantly and are in agreement with the content of the manuscript, and have approved it for submission. There are no potential competing interests.  

    We hope that the revised manuscript now is suitable for publication in Biomolecules. Thank you very much for your kind attention to our manuscript, and we look forward to hearing from you.  

Sincerely

Dong-Qiu Dai

Department of Surgical Oncology

The Fourth Affiliated Hospital

China Medical University

Shenyang 110032, China

Phone: +86-24-6204-3469

Fax: +86-24-6204-3110

Manuscript ID: biomolecules-2933045

Editor and Reviewers' Comments in Bold Letters  

Authors' replies in red

Revised sections of the manuscript are marked in red

Response to Reviewer 1 comments

In this comprehensive review, the author provides a thorough summary of the current knowledge surrounding long non-coding RNAs (lncRNAs) in drug resistance of gastric cancer (GC). This review aims to summarize the molecular mechanisms underlying GC drug resistance regulated by lncRNAs and also discuss the potential clinical applications of lncRNAs as biomarkers and therapeutic targets in GC. This article highlights the importance of understanding the role of lncRNAs in GC, particularly in the context of the Special Role of Drug-Resistant lncRNAs, CeRNA Model and Regulatory Mechanisms, Functional Significance in Tumor Microenvironment, Metabolism, and Drug Resistance, Utilization for Diagnosis and Treatment Guidance. However, a noteworthy reference is the insightful review article in Frontiers in Oncology (2022, 12: 841411), which focused on similar topics. To enhance the value of this review and contribute novel insights, the author should incorporate the following comments:

  1. Consider using bullet pointsto highlight key information in the simple summary for improved readability and comprehension.

Reply:Thank you very much for your suggestions on the revision of the manuscript.According to your modification requirements, we have highlighted the summary by using bullet points. The revised content is detailed in lines 17 to 29 of the document.

Changes in the text:Line 17-29

  1. Line 59: Replace "antioncogenes" with "tumor suppressor."

Reply:Thank the reviewers for their valuable comments.We have replaced the words according to your request (61 lines, the modified position).

Changes in the text:Line 61

  1. Consider whether the cartoon illustrating the gene motif is necessary for the review's content and purpose. If not, it may be omitted.

Reply:Thank you very much for your new understanding and revision suggestions on the content of the article. Your comments are very important to improve the treatment of the manuscript.We have deleted the gene part of Figure1 in accordance with your review requirements, and see the revised Figure1 for details.

Changes in the text:Line 1337(new Figure 1)

  1. Change the data series based on the resistance column. Replace "multi-drug" with specific drugs. Include HOTAIR and CAS9 involvement in Adriamycin resistance (Table 1).

Reply:Many thanks to the reviewers for their valuable advice on revising the article.As required, we have replaced all descriptions of "multi-drug" in Table 1 with specific drugs involved in the corresponding reference content, and the detailed modification has been reflected in Table 1, please review it again

Changes in the text:Line 1292

Provide more description about PD-1 and bevacizumab resistance, as these have not been discussed in the referenced article (Frontiers in Oncology).

Reply:Thank you very much for your valuable comments. Your comments can provide a strong guarantee for improving the quality and depth of the article.In this review, only two articles detailed how lncRNAs participate in the regulation of immune checkpoint inhibitor drug resistance in gastric cancer treatment. According to your comments, we will add more detailed instructions in the article, please refer to it. In addition, the involvement of lncRNA in the drug resistance of bevacizumab in gastric cancer is reflected in the conclusion and prospect section under the premise of ensuring logic.

Change in the text:Line 550-551,557-565,649-656.

  1. Modify the results series based on the Functions column for clarity and consistency (Table 2).

Reply:Thank you very much for your comments. Your comments have made the table more clear and precise.We have grouped Table 2 according to the function column section and redrawn a new Table 2 (which has been updated). At the same time, the corresponding function classification has been updated and explained in the corresponding result part of the article .

Changes in the text:line 1320;line 271-273.

6.Few abbreviations were missing in the text. Please provide abbreviations for DR (Drug Resistance) and FAQ in the figure legend for reader understanding (Fig. 2).

Reply:Thank the reviewers for their meticulous correction of the manuscript.Detailed explanations of the “DR” ,“DS”,“S/T” and “FAO” have been commented in the legend section of Figure2 and Figure4.

Changes in the text:Line 1342,line 1345,line 1375

7.Include each lncRNA mentioned, such as A9-ARlncRNA, in the table for completeness. Replace "ARlncRNAs" with "autophagy-resistant" and "SRlncRNAs" with "stemness-resistant" for clarity (Table 2).

Reply:Thank you very much for your professional review of the manuscript.For each lncRNA included in each model has been added to Table 3 as per your request. We try our best to show the names of all lncRNAs in Table 3, but only the ID of lncRNA is provided in the original data published in Reference 202. Therefore, the content provided in Reference 202 is presented in the form of lncRNA ID.

We also replace "ARlncRNAs" with "autophagy related lncRNAs", All changes are presented in a new Table 3 table.

Changes in the text:line 1331

  1. Correct all mistakes in the text, including ensuring consistent spacing between sentences.

Reply:Thank you for your meticulous review of the manuscript.We have We have carefully revised the formatting of all documents once again, including the spacing between sentences.

Reviewer 2 Report

Comments and Suggestions for Authors

Their appreciated authors, you did a great effort summarizing all the information about LncRNA, although you don't review the text. It contains writing issues such as: punctuation, spelling, grammar, etc .

The following observations are related more to the way in which the text is presented. It is necessary to review the presentation of the text.

When the acronyms of the word are indicated, do so by section and indicate them immediately afterwards. As they did in the Abstract for Gastric Cancer (GC). But in Simple Summary, they directly mention LncRNAs.

In line 62, please add references.

Please check the errors in the text such as line 290. (Figure 1). Change to (figure 1).

A table of abbreviations is necessary at the end of the article.

Comments on the Quality of English Language

Can the authors ask for special help in order for the text to be formal and easy to understand.

Author Response

March 25, 2024

Editor-in-Chief,

Biomolecules,  

Dear Editor,  

    Thank you very much for your letter and the reviewers’ very constructive comments. We recognized that you and the reviewers kindly spent a lot of time and provided expert comments.  

    In response to the comments, we have made point-by-point modifications to address all the comments, and the details of modifications are described in response letter to reviewers.

    This manuscript has not been published, and is not under consideration elsewhere.  

    All the authors have contributed significantly and are in agreement with the content of the manuscript, and have approved it for submission. There are no potential competing interests.  

    We hope that the revised manuscript now is suitable for publication in Biomolecules. Thank you very much for your kind attention to our manuscript, and we look forward to hearing from you.  

Sincerely

Dong-Qiu Dai

Department of Surgical Oncology

The Fourth Affiliated Hospital

China Medical University

Shenyang 110032, China

Phone: +86-24-6204-3469

Fax: +86-24-6204-3110

Manuscript ID: biomolecules-2933045

Editor and Reviewers' Comments in Bold Letters  

Authors' replies in red

Revised sections of the manuscript are marked in red

Response to Reviewer 2 comments

Their appreciated authors, you did a great effort summarizing all the information about LncRNA, although you don't review the text. It contains writing issues such as: punctuation, spelling, grammar, etc .

The following observations are related more to the way in which the text is presented. It is necessary to review the presentation of the text.

Reply:Thank the reviewers for taking the time to review the manuscript. Your valuable comments on the manuscript will be very important to improve the quality of the manuscript.

When the acronyms of the word are indicated, do so by section and indicate them immediately afterwards. As they did in the Abstract for Gastric Cancer (GC). But in Simple Summary, they directly mention LncRNAs.

Reply:Thank you very much for your review of the manuscript.We have checked the full text in view of the problems you mentioned, and revised the manuscript according to your requirements. See "Changes in the text" below for details.

Changes in the text:line 21,line 74;line 74-75;line 255;line 404-405;line 550-551;line 608;

In line 62, please add references.

Reply:Thank you very much for your meticulous review.We have added the corresponding reference at the end of line 63.

Changes in the text:Page 2,line 63.

Please check the errors in the text such as line 290. (Figure 1). Change to (figure 1).

Reply:Thank you very much for your meticulous review.All similar questions have been corrected

Changes in the text:line 115;line 158;line 185;line 249;line 266;line 333;line 366;line 383;line 399;line 448;line 495;line 539;line 573;line 591.

A table of abbreviations is necessary at the end of the article.

Reply:Thank you very much for your meticulous review.Your opinion is of great significance in improving the standard and scientific nature of the article.We have added the table at the end of the article as per your request.

Changes in the text:line 667.

Round 2

Reviewer 2 Report

Comments and Suggestions for Authors

Thank you very much authors for paying attention to the observations made. Your work will be very helpful to the scientific community.

It only remains for me to comment that a final review of the form of the text is necessary before it is published.

Author Response

May 4, 2024

Editor-in-Chief,

Biomolecules,  

Dear Editor,  

    Thank you very much for your letter and the reviewers’ very constructive comments. We recognized that you and the reviewers kindly spent a lot of time and provided expert comments.  

    In response to the comments, we have made point-by-point modifications to address all the comments, and the details of modifications are described in response letter to reviewers.

    This manuscript has not been published, and is not under consideration elsewhere.  

    All the authors have contributed significantly and are in agreement with the content of the manuscript, and have approved it for submission. There are no potential competing interests.  

    We hope that the revised manuscript now is suitable for publication in Biomolecules. Thank you very much for your kind attention to our manuscript, and we look forward to hearing from you.  

Sincerely

Dong-Qiu Dai

Department of Surgical Oncology

The Fourth Affiliated Hospital

China Medical University

Shenyang 110032, China

Phone: +86-24-6204-3469

Fax: +86-24-6204-3110

Manuscript ID: biomolecules-2933045

Editor and Reviewers' Comments in Bold Letters  

Authors' replies in red

Revised sections of the manuscript have traces of modification in the article

Response to Reviewer 2(Round 2) comments

Thank you very much authors for paying attention to the observations made. Your work will be very helpful to the scientific community.

It only remains for me to comment that a final review of the form of the text is necessary before it is published.

Reply:Thank you very much for your recognition and praise of our team's work. We are also very grateful for your valuable comments on the revision of the manuscript, your comments and suggestions have a great help to improve the quality of the article.According to your request, we have carefully reviewed all the contents of the article again, and corrected the spelling errors, formatting errors, sentences and spacing problems between words.
